# An mTORC1-mediated negative feedback loop constrains amino acid-induced FLCN-Rag activation in renal cells with TSC2 loss

Kaushal Asrani [1] ✉, Juhyung Woo[1], Adrianna A. Mendes[1], Ethan Schaffer [1], Thiago Vidotto[1], Clarence Rachel Villanueva[1], Kewen Feng[2], Lia Oliveira[1], Sanjana Murali[1], Hans B. Liu[1], Daniela C. Salles [1], Brandon Lam[1], Pedram Argani[1] & Tamara L. Lotan [1,2,3] ✉

The mechanistic target of rapamycin complex 1 (mTORC1) integrates inputs from growth factors and nutrients, but how mTORC1 autoregulates its activity remains unclear. The MiT/TFE transcription factors are phosphorylated and inactivated by mTORC1 following lysosomal recruitment by RagC/D GTPases in response to amino acid stimulation. We find that starvation-induced lysosomal localization of the RagC/D GAP complex, FLCN:FNIP2, is markedly impaired in a mTORC1-sensitive manner in renal cells with *TSC2* loss, resulting in unexpected TFEB hypophosphorylation and activation upon feeding. TFEB phosphorylation in *TSC2*-null renal cells is partially restored by destabilization of the lysosomal folliculin complex (LFC) induced by FLCN mutants and is fully rescued by forced lysosomal localization of the FLCN:FNIP2 dimer. Our data indicate that a negative feedback loop constrains amino acid-induced, FLCN:FNIP2-mediated RagC activity in renal cells with constitutive mTORC1 signaling, and the resulting MiT/TFE hyperactivation may drive oncogenesis with loss of the *TSC2* tumor suppressor.

The mechanistic target of rapamycin complex 1 (mTORC1) kinase complex is a central regulator of cellular growth and proliferation, activated in response to nutrients and growth factor stimuli[1]. In current models, mTORC1 kinase activity is tightly regulated by two classes of small GTPases: Rag GTPase heterodimers (RagA/B with RagC/D) and RHEB. In response to nutrients, Rags recruit the key mTORC1 subunit Raptor to the lysosome, where RHEB, an allosteric mTOR activator, resides[2]. In parallel, growth factor signaling leads to inactivation of the RHEB GTPase activating protein (GAP), TSC2, or PRAS40 with subsequent activation of mTORC1. The majority of studies examining mTORC1 activity have focused on two well-characterized substrates: eIF4E-binding protein 1 (4EBP1), which activates cap-dependent translation, and ribosomal S6 kinase 1 (S6K1), which promotes protein synthesis. These substrates are recruited to the lysosome and mTORC1 by binding to Raptor via a five amino-acid-conserved TOR signaling (TOS) motif, and their phosphorylation is the most commonly used readout of mTORC1 activity[3,4].

However, another key class of mTORC1 substrates are recruited to the lysosome exclusively by Rag GTPases, thus providing a readout of mTORC1 activation in response to nutrient availability. The Microphthalmia family (MiT/TFE) is comprised of four conserved basic leucine zipper transcription factors (*MITF/TFE3/TFEB/TFEC*) that regulate

[1]Department of Pathology, Johns Hopkins University School of Medicine, Baltimore, MD, USA. [2]Department of Oncology, Johns Hopkins University School of Medicine, Baltimore, MD, USA. [3]Department of Urology, Johns Hopkins University School of Medicine, Baltimore, MD, USA. ✉e-mail: kasrani1@jhmi.edu; tlotan1@jhmi.edu

expression of CLEAR (Coordinated Lysosomal Expression and Regulation) genes. Phosphorylation of TFEB or TFE3 by mTORC1 promotes interaction with 14-3-3 chaperones in the cytosol, and results in the cytoplasmic retention and proteasomal degradation of these transcription factors[5]. Accordingly, short-term pharmacological mTORC1 inactivation with the ATP-competitive inhibitor torin promotes MiT/TFE de-phosphorylation, nuclear localization and activity, although rapamycin does not (likely due to its incomplete allosteric inhibition of mTOR kinase)[6–12]. In contrast to other mTORC1 substrates, TFEB lacks a TOS motif and depends solely on amino-acid-stimulated RagC/D activation for lysosomal recruitment and phosphorylation by mTORC1[13]. Accordingly, loss of folliculin (FLCN; a GAP for RagC/D) decreases TFEB phosphorylation[13] and increases TFEB/TFE3 nuclear localization[9,13,14]. Taken together, these findings are consistent with the known role of mTORC1 in the negative regulation of catabolic processes such as autophagy when nutrients are replete[1].

However, a number of studies have emerged recently to indicate that TFEB regulation by mTORC1 is more complicated than previously appreciated. In previous work by our group and others, constitutive mTORC1 hyperactivity due to TSC1/2 loss paradoxically positively regulated TFEB-dependent lysosomal gene expression[15–17] and promoted MiT/TFE nuclear localization in an mTORC1-dependent manner[15–20]. Accordingly, in a study published while this manuscript was being drafted, hypophosphorylation of TFEB was confirmed in cells with TSC2 loss[17]. The relevance of these findings for human health is evidenced by molecular pathology data demonstrating that subsets of human tumors driven by TSC1/2 genomic alterations (including perivascular epithelioid cell tumors and eosinophilic renal cell carcinomas) may alternatively be driven by mutually exclusive TFEB/TFE3 gene rearrangements or FLCN loss leading to constitutive MiT/TFE activity[21–28]. Accordingly, MiT/TFE-target gene products, such as GPNMB protein, provide an excellent biomarker for tumors with TSC1/2 alterations or TFEB/TFE3 gene rearrangements, suggesting high MiT/TFE activity in both types[29].

Taken together, these data suggest that MiT/TFE may be a previously unappreciated driver of tumorigenesis in the setting of TSC1/2 loss. However, the mechanism by which mTORC1 hyperactivation might result in a paradoxical increase in MiT/TFE activity remains unclear. In the current study, we describe a negative feedback loop that constrains amino-acid-induced mTORC1 activity vis a vis TFEB, in the setting of TSC2 loss. We demonstrate that starvation-induced lysosomal localization of the FLCN:FNIP2 complex and downstream Rag activation in response to amino acids are markedly impaired in a rapamycin-sensitive manner in cells with TSC2 loss. Importantly, TFEB phosphorylation remains responsive to inputs that re-localize or destabilize the lysosomal folliculin complex (LFC). These findings begin to elucidate the mechanism of MiT/TFE hyperactivity in the context of TSC2 loss, which may be an important mediator of tumorigenesis.

## Results
### mTORC1 hyperactivation via TSC2 loss drives lysosomal biogenesis in murine and human renal tumor cells
We previously demonstrated that mice with constitutive epidermal mTORC1 activity (via conditional epidermal deletion of Tsc1 or expression of constitutively active RhebS16H) exhibited elevated MiT/TFE transcriptional activity, with concomitantly increased expression of CLEAR target genes and proteins[16]. To determine whether these paradoxical findings in primary keratinocytes might be generalizable to other systems, we examined transformed human embryonic kidney HEK293T cells with or without somatic genomic deletion (KO) of TSC1, TSC2 or TSC1/2 via CRISPR-Cas9 genome editing (gift of TSC Alliance and Dr. Nellist[30]) (Fig. 1a), (hereafter referred to as WT and TSC1, TSC2, or TSC1,2 KO cells, respectively). By RNA-seq, TFEB-regulated[31,32] lysosomal gene sets were enriched by GSEA in the TSC1,2 KO cells

compared to WT controls (Fig. 1b). Multiple CLEAR-regulated transcripts were significantly enriched in TSC2 and TSC1,2 KO cells, compared to WT cells by RT-PCR, with similar, though non-significant, increases in TSC1 KO cells (Fig. 1c, Fig. S1a). Validating the gene expression findings, multiple lysosomal integral proteins and enzymes were upregulated in whole-cell extracts of TSC2 KO and TSC1,2 KO cells, and in lysosomal-enriched fractions of TSC2 KO cells by immunoblotting (Fig. 1d, e). The lysosomes in TSC2 KO cells were functionally active, as measured indirectly by an increase in cathepsin B (CTSB) processing (Fig. 1d; arrow), as well as increased lipidated LC3-II (Fig. 1d, e; arrow), increased LC3-labeled puncta by immunofluorescence (Fig. S1b), and higher autophagic flux demonstrated by increased LC3-II accumulation (arrow) with hydroxychloroquine (HCQ) and Chloroquine (CQ) treatment[33] (Fig. S1c). Finally, there was a significant increase in Cathepsin D enzyme activity in TSC2 KO cells compared to WT (Fig. S1d).

To validate these findings in vivo, we also examined lysosomal biogenesis in Tsc2 ± A/J mice. In this murine model of tuberous sclerosis, spontaneous loss of the second Tsc2 + allele results in Tsc2 protein loss[29] and development of mTORC1-driven renal cystadenomas and cystadenocarcinomas at 6–12 months of age with 100% penetrance[34]. We performed laser-capture microdissection (LCM) on renal tumors from Tsc2 ± mice and found that levels of CLEAR transcripts were significantly enriched in tumors compared to matched normal kidney by RT-PCR (Fig. 1f, g) and were validated by immunohistochemistry (IHC)/immunofluorescence (IF) (Fig. 1h). For additional validation, we examined TTJ cells, a Tsc2-null cell line derived from Tsc2 ± C57BL/6 mice, stably transfected with a control vector (TTJ-parental) or wild-type Tsc2 (TTJ-Tsc2)[35]. TTJ-parental cells had increased lysosome/autophagosome proteins in whole-cell lysates and in lysosomal fractions by immunoblotting when compared to cells with Tsc2 re-expression. Notably, these cells also had increased processed Cathepsin B and lipidated LC3-II, consistent with a functional increase in lysosomal proteolysis and autophagic flux, respectively (Fig. 1i, j; arrow).

### MiT/TFE nuclear localization and transcriptional activity is increased in vitro with TSC2 loss
To understand the physiological basis for elevated lysosomal content and activity in TSC2 KO cells, we first examined the subcellular localization of TFEB and TFE3, the principal drivers of autophagy and lysosomal biogenesis in mammalian cells. Paradoxically, but in accordance with our previous findings in Tsc1 KO keratinocytes[16], nuclear localization of endogenous TFEB and TFE3 was increased in TSC2 KO cells compared to WT controls in nutrient-replete conditions (Fig. 2a, b; Fig. S2a, b). Furthermore, transiently overexpressed GFP-tagged TFEB (TFEB-GFP) (Fig. 2c, d) or TFE3 (TFE3-GFP) (Fig. S2c, d) showed increased nuclear: cytoplasmic localization in TSC2 KO cells versus WT controls. Immunoblotting of nuclear/cytoplasmic fractions confirmed these findings for endogenous TFEB and TFE3 (Fig. 2e–g, Fig. S2e) and exogenous TFEB-GFP (Fig. S2f). Allosteric mTORC1 inhibition with rapamycin significantly suppressed the nuclear localization of both TFEB and TFE3, while the ATP-competitive dual mTORC1/2 kinase inhibitor torin did not significantly affect TFEB localization though it did suppress TFE3 nuclear localization (Fig. 2e–g). The increased TFEB and TFE3 nuclear localization in TSC2 KO cells resulted in a significant increase in MiT/TFE transcriptional activity, as measured by 4XCLEAR promoter (containing 4 tandem copies of a CLEAR promoter element), luciferase reporter assays[36] (Fig. 2h), These effects were confirmed in vivo using TSC2 KO cells grown as allogenic xenografts in NSG (NOD scid gamma) mice where there was elevated TFEB and TFE3 nuclear localization compared to WT controls by genetically validated immunohistochemistry (Fig. 2i, j). We then examined tumor xenograft growth

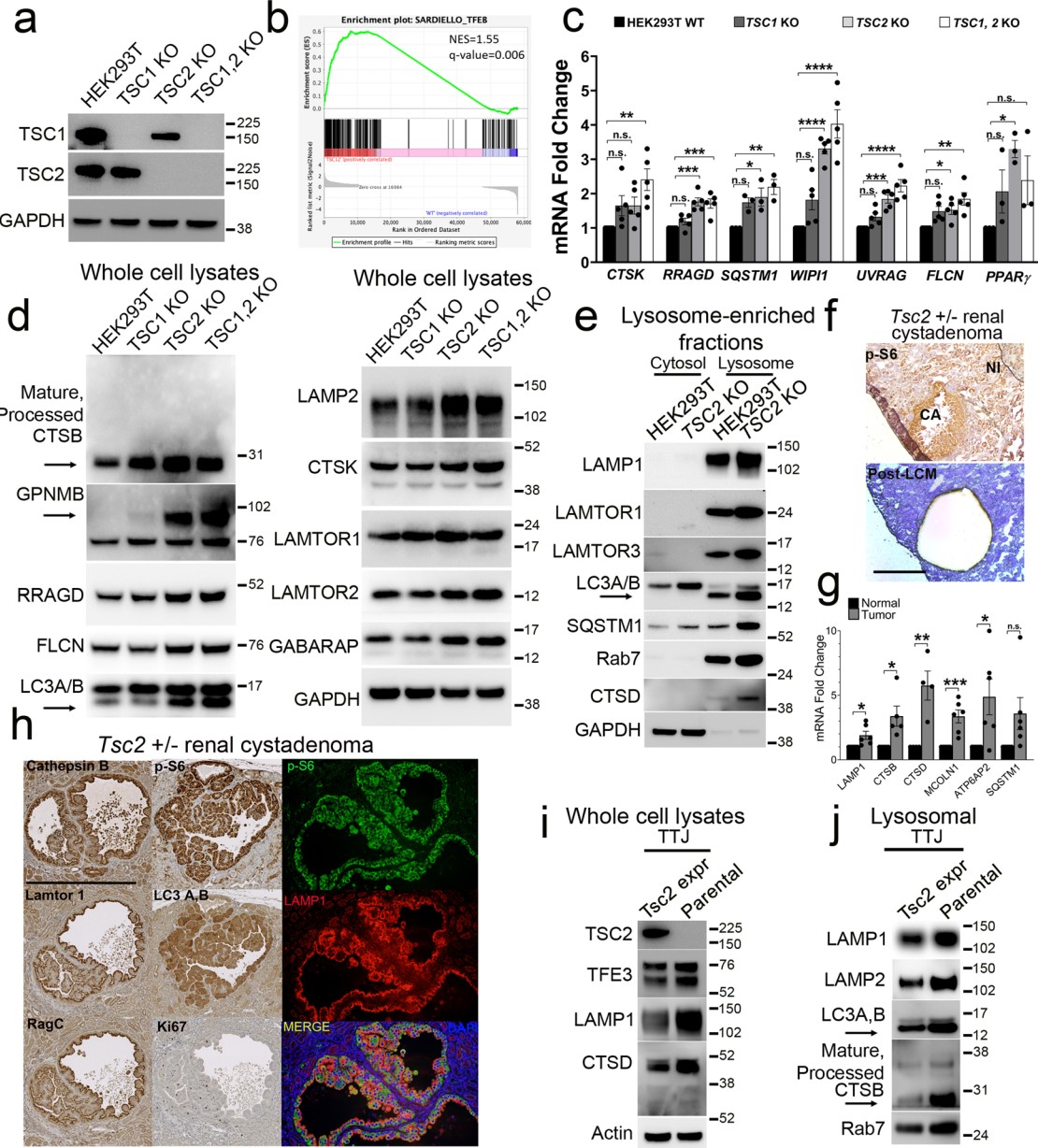

**Fig. 1 | mTORC1 hyperactivation via TSC2 loss drives lysosomal biogenesis in murine and human renal tumor cells. a** Immunoblotting of HEK293T WT cells and cells with CRISPR-Cas9-mediated genomic deletion of *TSC1*, *TSC2*, and *TSC1/2*. **b** Gene Set Enrichment Analysis (GSEA) comparing *TSC1/2* KO versus WT cells for TFEB transcriptional targets[31]. **c** Quantitative real-time PCR (qRT-PCR) for lysosomal CLEAR target gene transcripts in *TSC1*, *TSC2*, and *TSC1/2* KO cells compared to WT controls. n = 3 *(SQSTM1, PPARγ)* and 5 *(CTSK, RRAGD, WIPI1, UVRAG, FLCN)* independent biological replicates. Graphs are presented as mean values ± SEM. Statistical analyses were performed using one-way ANOVA with Dunnett's test for multiple comparisons. *p < 0.05, **p < 0.01, ***p < 0.001, ****p < 0.0001 (also see Fig. S1a). Immunoblotting for lysosomal proteins in: **d** whole-cell lysates of *TSC1*, *TSC2*, and *TSC1/2* KO cells compared to WT controls and **e** lysosomal fractionation assays of *TSC2* KO cells, compared to WT cells. Arrows in **d** indicate increased mature, processed Cathepsin B and a specific band for GPNMB and arrows in

**d**, **e** indicates lipidated LC3-II accumulation in *TSC2* KO cells. **f** Representative tumor sections immunostained for p-S6 (top) and hematoxylin-stained slides following laser-capture microdissection of murine *Tsc2* ± renal cystadenomas (bottom) (Scale bar = 500 µm). **g** qRT-PCR from micro-dissected tumor samples in **f** for lysosomal genes compared to matched normal kidney. n = 4 *(CTSD)*, 5 *(CTSB)*, and 6 *(LAMP1, MCOLN1, ATP6AP2,* and *SQSTM1)* independent biological replicates. Graphs are presented as mean values ± SEM. Statistical analyses were performed using two-tailed Students *t*-test. *p < 0.05, **p < 0.01, ***p < 0.001. **h** Immunostaining of *Tsc2* ± renal cystadenomas for lysosomal proteins. (Scale bar = 1 mm). Immunoblotting of *Tsc2⁻/⁻* TTJ cells for lysosomal proteins in **i** whole-cell lysates and **j** lysosomal fractions compared to TTJ cells with Tsc2 re-expression. Processed CTSB (arrow) and LC3-II (arrows) are indicated. Source data are provided as a Source Data file.

in WT and *TSC2* KO cells in response to treatment with rapamycin and torin. Rapamycin significantly decreased tumor growth in both WT and *TSC2* KO xenografts (Fig. 2k, Figs. S3a, S3b) consistent with decreased TFEB nuclear localization. In contrast, torin did not significantly affect tumor growth (Fig. 2k), consistent with the lack of effect on TFEB nuclear localization, despite suppressed

phosphorylation of classic mTORC1 substrates (p-4E-BP1 and p-p70S6K) in *TSC2* KO tumor lysates (Fig. 2l and Fig. S3a, b). Finally, *Tsc2*-null TTJ-parental cells showed an increase in nuclear: cytoplasmic TFEB in nuclear-fraction immunoblots, compared to *Tsc2* re-expressing cells (Fig. S2g), thus validating our findings in an orthogonal renal tumor model system.

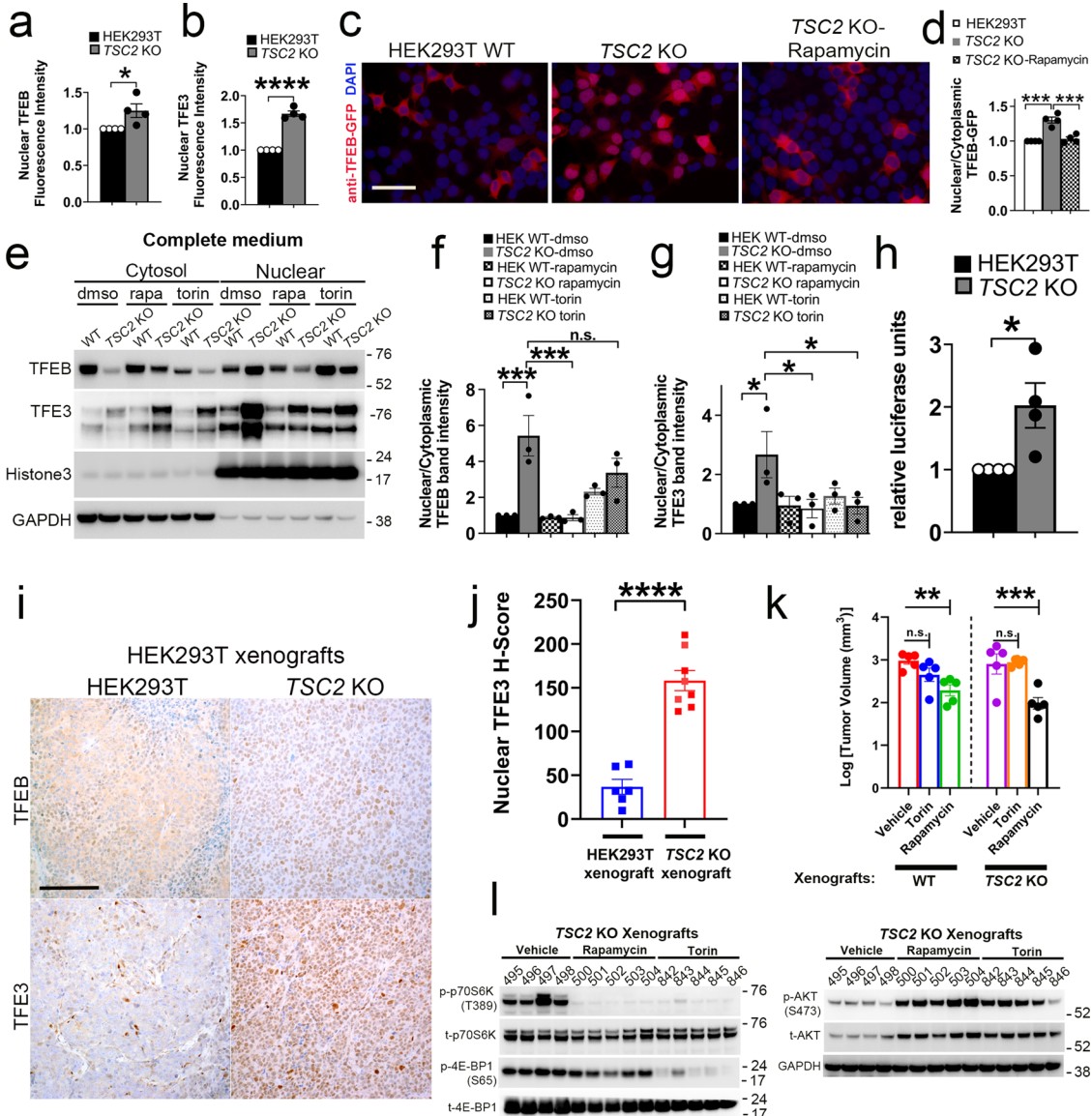

**Fig. 2 | MiT/TFE nuclear localization and transcriptional activity is increased in vitro with TSC2 loss.** Indirect immunofluorescence quantification for TFEB (**a**), and TFE3 (**b**) in *TSC2* KO cells compared to WT controls (from experiments in Fig. S2a and S2b, respectively). *n* = 4 independent biological replicates, counting 2565 cells (for TFE3) and 780 cells (for TFEB), per replicate. *p* = 0.0369 for TFEB and *p* < 0.0001 for TFE3. **c** Indirect immunofluorescence for GFP in WT and *TSC2* KO cells transiently transfected with TFEB-GFP for 24 h ± rapamycin (200 nM, 2 h) (Scale bar = 100 μm). **d** Quantification of nuclear/cytoplasmic TFEB-GFP fluorescence from experiments in **c**. *n* = 4 independent biological replicates, counting 150 cells per replicate. *p* = 0.0001 (WT vs *TSC2* KO) and *p* = 0.0002 (*TSC2* KO ± rapamycin). **e** TFEB and TFE3 expression in nuclear-fraction immunoblots of *TSC2* KO cells compared to WT controls ± rapamycin (200 nM, 2 h) or torin (1 μM, 2 h). **f, g** Densitometry quantification of immune-blot experiments from **e**. *n* = 3 independent biological replicates. \**p* < 0.05, \*\*\**p* < 0.001. **h** 4X-CLEAR luciferase activity in *TSC2* KO cells compared to WT controls, normalized to Renilla luciferase. *n* = 4

independent biological replicates. *p* = 0.0282. **i** Representative TFEB and TFE3 immunohistochemistry images for *TSC2* KO xenografts in NSG mice, compared to WT xenografts (Scale bar = 200 μm). **j** Digital quantification of mean nuclear TFE3 H-score from experiments in **i**. *n* = 6 (WT xenografts) and 8 (*TSC2* KO xenografts) independent biological replicates. *p* < 0.0001. **k** In vivo growth and tumor volume of subcutaneous WT and *TSC2* KO xenografts in NSG mice treated with vehicle, rapamycin or torin. *n* = 5 independent biological replicates. *p* = 0.0092 (rapamycin; WT xenografts) and *p* = 0.0005 (rapamycin; *TSC2* KO xenografts).
**l** Immunoblotting of tumor lysates from *TSC2* KO xenografts (from experiments in **k**) treated with vehicle, rapamycin or torin (also see Fig. S3b). All graphs are presented as mean values ± SEM. Statistical analyses were performed using two-tailed Students *t*-test (panels **a**, **b**, **h**, and **j**) or one-way ANOVA with Dunnett's test for multiple comparisons (panels **d**, **f**, and **g**) or Bonferroni's test for multiple comparisons (panel **k**), if more than 2 groups. Source data are provided as a Source Data file.

## TFEB and TFE3 have increased nuclear localization in murine renal tumors associated with Tsc2 loss in vivo compared to normal kidney

We next examined murine tumors with constitutive mTORC1 activity to determine MiT/TFE nuclear localization in vivo. In murine renal cystadenomas with *Tsc2* loss, nuclear localization of TFEB and TFE3 was increased compared to surrounding normal renal tubules (Fig. 3a), consistent with the previously observed

increase in MiT/TFE transcriptional activity (Fig. 1f, g) and this difference was statistically significant on digital quantification (Fig. 3b). We examined two additional mouse models of conditional *Tsc1/2* loss-induced renal tumorigenesis: (a) renal cystadenomas from *Pax8 Cre;Tsc2*fl/wt mice, where Tsc2 protein loss was also confirmed, consistent with spontaneous bi-allelic inactivation of *Tsc2* (Fig. 3c), and (b) *Tsc1*−/− renal cystadenomas from *Rosa(ER) Cre;Tsc1*fl/fl mouse treated with 4-OHT (Fig. 3d). Both these mice

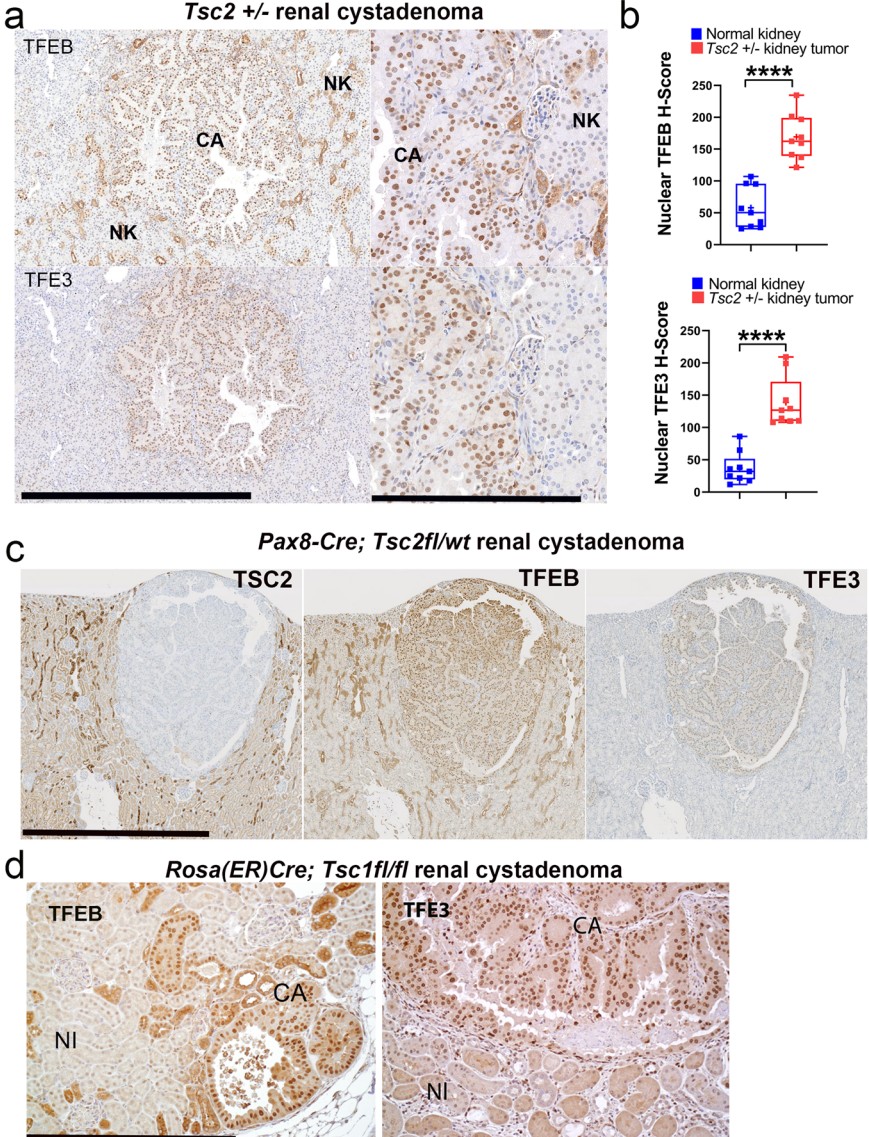

**Fig. 3 | TFEB and TFE3 have increased nuclear localization in murine renal tumors associated with TSC2 loss in vivo compared to normal kidney.**
**a** Representative immunohistochemistry (IHC) for TFEB (top row) and TFE3 (bottom row) in *Tsc2*± murine renal cystadenomas (CA), compared to surrounding normal kidney (NK) (Scale bar = 1 mm; inset = 250 μm). **b** Digital quantification of mean nuclear TFEB and TFE3 H-scores from experiments in **a**, as depicted by box-and-whisker plots. All box plots include the central median line, "+" at the mean, the box denoting the interquartile range (IQR; extending from 25th to 75th percentile) with whiskers plotted down to the minimum and up to the maximum values. *n* = 9 independent biological replicates. Statistical analyses were performed using two-tailed Students *t*-test. *p* < 0.0001 (for TFEB and TFE3). **c** Representative IHC for TSC2 (left), TFEB (middle), and TFE3 (right) in renal cystadenomas from Pax8 Cre; Tsc2[fl/wt] mice at 18 months (Scale bar = 1 mm). **d** Representative IHC for TFEB (left) and TFE3 (right) in Tsc1[−/−] renal cystadenomas from Rosa(ER)Cre;Tsc1[fl/fl] mice treated with 4-OHT (Scale bar = 250 μm). Source data are provided as a Source Data file.

models also showed similarly elevated TFEB/ TFE3 nuclear localization.

## Genomic inactivation of TFEB and TFE3 decreases proliferation of TSC2 KO cells and xenografts

We next directly examined whether elevated MiT/TFE transcriptional activity drives lysosomal biogenesis and/or increases the proliferation of *TSC1/2*-null cells and tumors. We used CRISPR-Cas9 genome editing to knockout *TFEB, TFE3*, or both in *TSC2* KO cells, with cells expressing nontargeting gRNA as controls. In *TSC2* KO cells, knockout of *TFE3* (but not *TFEB*) was sufficient to reduce LAMP1 expression (Fig. 4a). Importantly, inactivation of *TFEB, TFE3*, or both, decreased the phosphorylation of mTORC1 substrates S6K and 4E-BP1, similar to what has been reported in *TSC2*-intact systems[37]. Simultaneous knockout of

both *TFEB* and *TFE3* (but not either gene alone) consistently decreased the proliferation of the *TSC2* KO clones in in vitro confluence and viability assays (Fig. S4a, b), and also reduced in vivo growth of subcutaneous xenografts (Fig. 4b, c, Fig. S5), consistent with a role for MiT/TFE proteins as potential drivers of tumor progression in cells with constitutive mTORC1 activation.

The results of the xenograft experiments suggested that *TFEB* and *TFE3* could partially compensate for one another in the context of *TSC2* loss, consistent with findings that have been reported in other systems in a context-specific manner[38,39]. In order to further characterize potential functional redundancy of *TFEB* and *TFE3* in the *TSC2* KO background, we performed RNA-seq on tumor xenografts grown from WT, *TSC2* KO, and *TFEB* KO, *TFE3* KO and double *TFEB/TFE3* KO cells (Fig. 4d). Multiple *TFEB*- and/or *TFE3*-regulated lysosomal gene

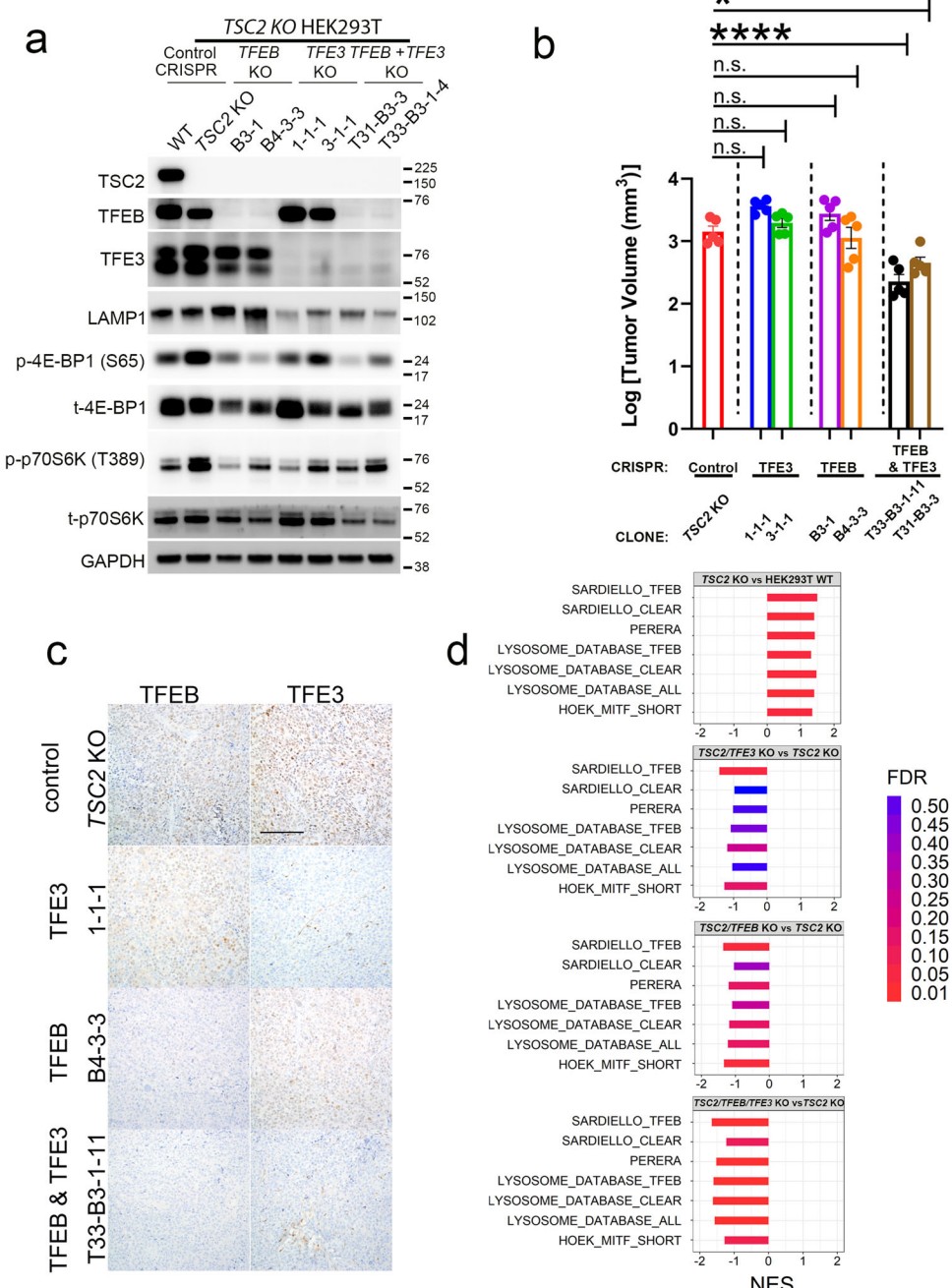

**Fig. 4 | Genomic inactivation of TFEB and TFE3 decreases proliferation of TSC2 KO cells and xenografts. a** Immunoblotting of *TSC2* KO control cells and cells with CRISPR-Cas9-mediated genomic inactivation of *TFEB, TFE3*, and dual inactivation of *TFEB and TFE3*, for TFEB, TFE3, LAMP1, and mTORC1 substrates (p-4EBP1 (S65) and p-p70 S6K (T389)). Two clones representing two unique guide RNAs targeting *TFEB* and *TFE3* are shown. **b** In vivo growth and tumor volume of subcutaneous *TSC2* KO xenografts in NSG mice with single or combined genomic deletion of *TFEB* and *TFE3*. *n* = 5 independent biological replicates. Graphs are presented as mean values ± SEM. Statistical analyses were performed using one-way ANOVA with

Dunnett's test for multiple comparisons. *p* < 0.0001 (*TSC2* KO vs T33-B3-1-11) and *p* = 0.0112 (*TSC2* KO vs T31-B3-3). **c** Representative TFEB and TFE3 immunohistochemistry in *TSC2* KO xenografts in NSG mice with single or combined genomic deletion of *TFEB* and *TFE3* (Scale bar = 100 μm). **d** Gene Set Enrichment Analysis (GSEA) comparing *TSC2* KO versus WT xenografts (top panel), *TSC2/TFE3* KO xenografts vs *TSC2* KO xenografts (2nd panel), *TSC2/TFEB* KO xenografts vs *TSC2* KO xenografts (3rd panel), and *TSC2/TFEB/TFE3* KO xenografts vs *TSC2* KO xenografts (4th panel), for TFEB/TFE3 transcriptional targets[31,40–42]. Source data are provided as a Source Data file.

sets[31,40–42] were positively enriched by GSEA in *TSC2* KO compared to WT xenografts, validating our initial in vitro findings (Fig. 1b). While *TFEB/TFE3* double KO xenografts in the *TSC2* KO background showed significant negative enrichment of all lysosomal gene sets compared to *TSC2* KO xenografts, this effect was partially attenuated by *TFEB* or *TFE3* KO alone. These results are largely in line with observed effects on tumor growth.

In contrast to the apparent redundancy of *TFEB* and *TFE3* for tumor growth in vivo, in vitro results (Fig. 4a) suggested that single *TFEB* or *TFE3* KO may be sufficient to dampen the phosphorylation of direct mTORC1 substrates (p-4E-BP1 and p-p70S6K). Notably, a prior study[37] demonstrated that MiT/TFE activation and CLEAR-mediated gene transcription may lead to increased RagD expression (since *RRAGD* contains a CLEAR element in its promoter) and thus contribute

to increased mTORC1 signaling. We examined *TSC2* KO cells and xenografts and observed that, consistent with our finding of CLEAR activation in these cells, *RRAGD* mRNA and RagD protein levels were increased in *TSC2* KO and *TSC1,2* KO cells and xenografts, compared to their WT counterparts (Fig. 1c, d and Fig. S6a, b). In *TSC2* KO cells, compared to scrambled control shRNA-transfected cells, *RRAGD* shRNA partially suppressed canonical mTORC1 substrate phosphorylation by immunoblotting (Fig. S6c). Significantly, single *TFE3* or *TFEB* KO or double *TFEB/TFE3* KO (Fig. S6d), suppressed *RRAGD* transcript levels in *TSC2* KO cells, paralleling the diminished mTORC1 signaling in vitro (Fig. 4a). These results indicate that similar to previously reported findings in WT cells, increased *RRAGD* transcription downstream of MiT/TFE activity may contribute to mTORC1 signaling in *TSC2* KO cells, and the resulting downregulation in *RRAGD* transcription in the *TFEB* and *TFE3* KO clones may be sufficient to partially dampen mTORC1 signaling in vitro. Taken together, our results suggest that MiT/TFE factors have partially redundant functions in a context-dependent and readout-dependent fashion in *TSC2* KO cells.

## TSC2 loss decreases phosphorylation of TFEB at canonical mTORC1 sites in an amino-acid-dependent and rapamycin-sensitive manner

MiT/TFE subcellular localization is primarily regulated by serine phosphorylation mediated by mTORC1 (at S211 and S122 in TFEB and S321 in TFE3), resulting in 14-3-3 binding and cytoplasmic sequestration[6–9]. TFEB phosphorylation occurs downstream of nutrient inputs to mTORC1, at S211 and S122[6–8,12] (in response to amino acids) and at S122[12] (in response to serum). Unlike other mTORC1 substrates, TFEB phosphorylation is insensitive to growth factor signaling, does not require RHEB, nor is it increased upon short-term *TSC2* knockdown[13]. Indeed, TFEB phosphorylation was responsive to amino acids and unaltered with glucose or serum addback in WT cells (Fig. S7a). Paradoxically, however, we found that endogenous p-TFEB (S211) and p-TFEB (S122) levels were markedly *decreased* in *TSC2* KO cells, comparable to levels seen with amino-acid starvation (Fig. 5a, b) or mTOR kinase inhibition in WT cells (Fig. 5c). Though TFEB phosphorylation in *TSC2* KO cells was unresponsive to exogenous amino-acid addback in the starved state (Fig. 5a, b, Fig. S7a), it did respond to cycloheximide (CHX) treatment in the fed state (Fig. S7b), which increases intra-cellular amino-acid levels by decreasing amino-acid utilization in protein synthesis[2,43].

TFEB hypophosphorylation was additionally observed in vivo in *TSC2* KO tumor xenografts (Fig. 5d), *Tsc2*-deficient, TTJ-parental cells (Fig. S7c), and *TSC2*-deficient human TRI-102 cells (an E6/E7 and hTERT- immortalized derivative of human *TSC2*-null, primary renal angiomyolipoma cells[44],) (Fig. S7d), compared to their *TSC2/ Tsc2* expressing, wild-type counterparts, respectively. GFP-tagged, exogeneous TFEB was also hypophosphorylated in *TSC2* KO and *TSC1/2* KO cells compared to WT (Fig. S7e). TFEB hypophosphorylation was reversed by rapamycin but not torin in *TSC2* KO cells (Fig. 5a–c) and xenografts (Fig. S3b), consistent with our findings for TFEB nuclear localization (Fig. 2c–g). Accordingly, immunoprecipitation experiments confirmed decreased interaction of TFEB with cytosolic 14-3-3 proteins in *TSC2* KO cells, and there was reversal of this finding with rapamycin but not torin (Fig. 5e, Fig. S7f). Cumulatively, these results indicate that increased TFEB/TFE3 nuclear localization in *TSC2* KO cells is due to TFEB hypophosphorylation at mTORC1-sensitive phosphosites and is paradoxically reversible by treatment with the allosteric mTOR inhibitor, rapamycin.

## Active RagC/D heterodimers rescue TFEB lysosomal localization, phosphorylation, and cytoplasmic retention in cells with TSC2 loss

Both TFEB[45,46] and Raptor[2] are recruited to the lysosome via active Rag GTPases in response to amino-acid stimulation. TFEB

hypophosphorylation was reversible with rapamycin treatment (Fig. 5a–c) consistent with the fact that lysosomal recruitment of TFEB[7,8,45] and Raptor are both increased by mTOR inhibition[47]. Thus, we hypothesized that RagC/D-mediated recruitment of TFEB to the lysosome might be impaired in cells with *TSC2* loss via a mechanism mediated by rapamycin-sensitive mTORC1 activity. By immunoprecipitation, the interaction of TFEB-GFP with endogenous RagA, RagC, and Raptor was decreased in *TSC2* KO compared to WT cells, and this was reversible with rapamycin (Fig. 5e). To begin to test our hypothesis that Rag activity might be constrained by an mTORC1-mediated mechanism in *TSC2* KO cells, we first examined lysosomal fractions by immunoblotting. Both TFEB and Raptor levels were decreased at the lysosome in *TSC2* KO compared to WT cells, and levels were increased upon mTORC1 inhibition with either rapamycin or torin (Fig. 6a, Fig. S8a). Rag GTPases cycle on and off the lysosome rapidly in response to nutrients, in a manner governed by their activation status[2,45,46]. In WT cells, lysosomal localization of RagC was strongly increased with amino-acid starvation and decreased by amino-acid addback, as previously described (Fig. S8a)[48]. However, this response was significantly blunted in *TSC2* KO cells and reversed with rapamycin, consistent with a defect in RagC activation.

Next, we examined whether expression of active Rag mutants, many of which are also tethered to the lysosome[2], might rescue TFEB phosphorylation in *TSC2* KO cells. Strikingly, expression of active RagC[GDP] (*RAGC* S75L) or active RagD[GDP] (*RAGD* S77L), but not inactive RagC[GTP] (*RAGC* Q120L), completely rescued TFEB phosphorylation at S211 and S122 in *TSC2* KO cells, without substantially increasing S6 phosphorylation in these cells (Fig. 6b, c), and also promoted TFEB phosphorylation in WT cells as expected (Fig. S8b). Interestingly, overexpression of active RagD further boosted 4EBP1 phosphorylation in *TSC2* KO cells (Fig. 6b), consistent with its role as a driver of canonical mTORC1 substrate phosphorylation. Overexpression of active RagB[GTP] (*RAGB* Q99L) in *TSC2* KO cells, also promoted partial rescue of TFEB phosphorylation, albeit to a lesser extent than active RagC (Fig. S8c), but co-expression of active RagB/C heterodimers was not additive over active RagC alone (Fig. S8c). Moreover, co-expression of inactive RagC[GTP] with active RagB[GTP] failed to stimulate TFEB phosphorylation, while co-expression of inactive RagB[GDP] with active RagC[GDP] still promoted significant TFEB phosphorylation in *TSC2* KO cells (Fig. S8c). Taken together, these data underscore the specific dependence of TFEB phosphorylation on RagC/D activation in *TSC2* KO cells, as previously described for wild-type cells[13]. Finally, both active RagC and D resulted in cytosolic relocalization of TFEB in cells with *TSC2* loss (Fig. 6d).

Since active Rag mutants were sufficient to rescue TFEB phosphorylation in *TSC2* KO cells, we next asked whether Rag-mediated lysosomal recruitment was required for the observed rescue of TFEB phosphorylation in the context of mTORC1 inhibition with rapamycin. The N-terminal domain of TFEB is essential for Rag binding and lysosomal recruitment[6,13,45,49]. Accordingly, TFEB mutants a) that lack the first 30 amino acids (Δ30-TFEB-GFP), or b) contain point mutations within the first 30 residues (TFEB-S3A, R4A-GFP), do not bind Rags, fail to relocate to the lysosome on mTORC1 inactivation and are constitutively nuclear localized[6,13,45,49]. As expected, Δ30-TFEB and S3A, R4A-TFEB were hypophosphorylated at S211 in WT cells, compared to GFP-WT TFEB[6,13,45,49] (Fig. 6e), and we also found TFEB to be hypophosphorylated at S122 in these cells. Similar to endogenous TFEB, rapamycin increased p-TFEB S211 and S122 on GFP-WT TFEB in *TSC2* KO cells, but this effect was not seen for Δ30-TFEB and S3A, R4A-TFEB, indicating that the N-terminal Rag-binding domain is essential for rapamycin-induced TFEB phosphorylation in the context of *TSC2* loss.

TFEB is an atypical mTORC1 substrate that lacks the conventional TOR signaling (TOS) motif known to mediate mTORC1 substrate recruitment via direct binding to Raptor[3,4], and

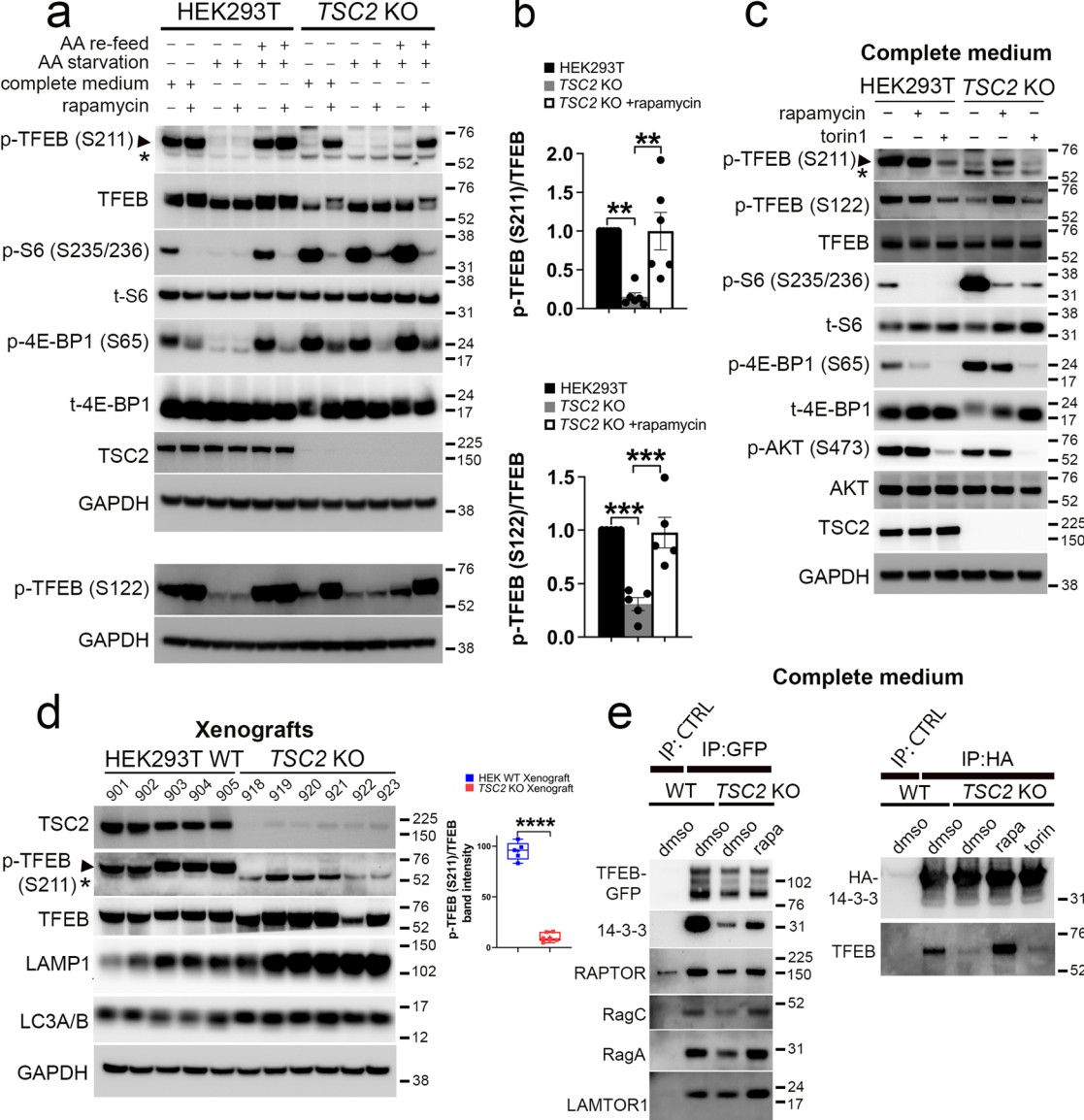

**Fig. 5 | TSC2 loss decreases phosphorylation of TFEB at canonical mTORC1 sites in an amino-acid-dependent and rapamycin-sensitive manner.**
**a** Immunoblotting of whole-cell lysates from WT and *TSC2* KO cells maintained in complete growth media (lanes 1,2,7,8), or following amino-acid starvation for 90 min (lanes 3,4,9,10), or starvation and restimulation with amino acids for 30 min (lanes 5,6,11,12). Cells were either treated with vehicle control (lanes 1,3,5,7,9,11) or rapamycin (200 nM; 2 h) (lanes 2,4,6,8,10,12). p-TFEB (S122) and GAPDH are non-contemporaneous immunoblots from the same biological replicate. Asterisk indicates a non-specific band. **b** Densitometry quantification of endogenous p-TFEB/TFEB ratios in WT and vehicle or rapamycin-treated *TSC2* KO cells from experiments in **a**. $n = 6$ (p-TFEB (S211)) and 5 (p-TFEB (S122)) independent biological replicates. Graphs are presented as mean values ± SEM. Statistical analyses were performed using one-way ANOVA with Dunnett's test for multiple comparisons. $**p < 0.01$, $***p < 0.001$. **c** Immunoblotting of whole-cell lysates from WT and TSC2 KO cells. Cells were treated with rapamycin (200 nM) or torin (1 μM) for 2 h prior to

lysis. **d** Immunoblotting of lysed WT and TSC2 KO xenografts. Densitometry quantification of mean p-TFEB (S211)/ TFEB ratios in WT and TSC2 KO xenografts on right. $n = 5$ (WT) and 6 (TSC2 KO) independent biological replicates. Box plots include the central median line, "+" at the mean, the box denoting the interquartile range (IQR; extending from 25th to 75th percentile) with whiskers plotted down to the minimum and up to the maximum values. Statistical analyses were performed using two-tailed Students $t$-test. $p < 0.0001$. **e** WT and TSC2 KO cells were transiently transfected with TFEB-GFP (24 h), followed by lysis, immunoprecipitation (IP) with either a binding control or a GFP-Trap coupled to magnetic agarose beads and immunoblotting (left panels). WT and TSC2 KO cells were transiently transfected with HA-tagged 14-3-3 gamma (24 h), followed by lysis, immunoprecipitation using an anti-HA antibody or control IgG and immunoblotting (right panels). TSC2 KO cells were treated with vehicle, rapamycin (200 nM; 2 h) or or torin (1 μM; 2 h), prior to lysis (also see Fig. S7f). Source data are provided as a Source Data file.

instead relies on the Rag GTPases for lysosomal recruitment[13]. To bypass the requirement of RagC/D for TFEB recruitment in *TSC2* KO cells, we leveraged a previously described TFEB substitution-mutant chimera wherein the first 30 amino acids of TFEB are replaced with the first 30 amino acids of S6K containing the TOS motif (GFP-TOS-Δ30-TFEB)[13]. In contrast to GFP-WT TFEB, there were similarly high levels of phosphorylation of GFP-TOS-Δ30-TFEB in *TSC2* KO

compared to WT cells. In contrast, TOS-(F5A)-Δ30-TFEB-GFP (a variant of TOS-Δ30-TFEB-GFP in which a key phenylalanine residue of the TOS motif has been mutagenized to alanine), was equally hypophosphorylated in both WT and *TSC2* KO cells (Fig. 6f). Cumulatively, these data indicate that lysosomal recruitment and phosphorylation of TFEB are constrained in a RagC/D- and mTORC1-dependent manner in *TSC2* KO cells.

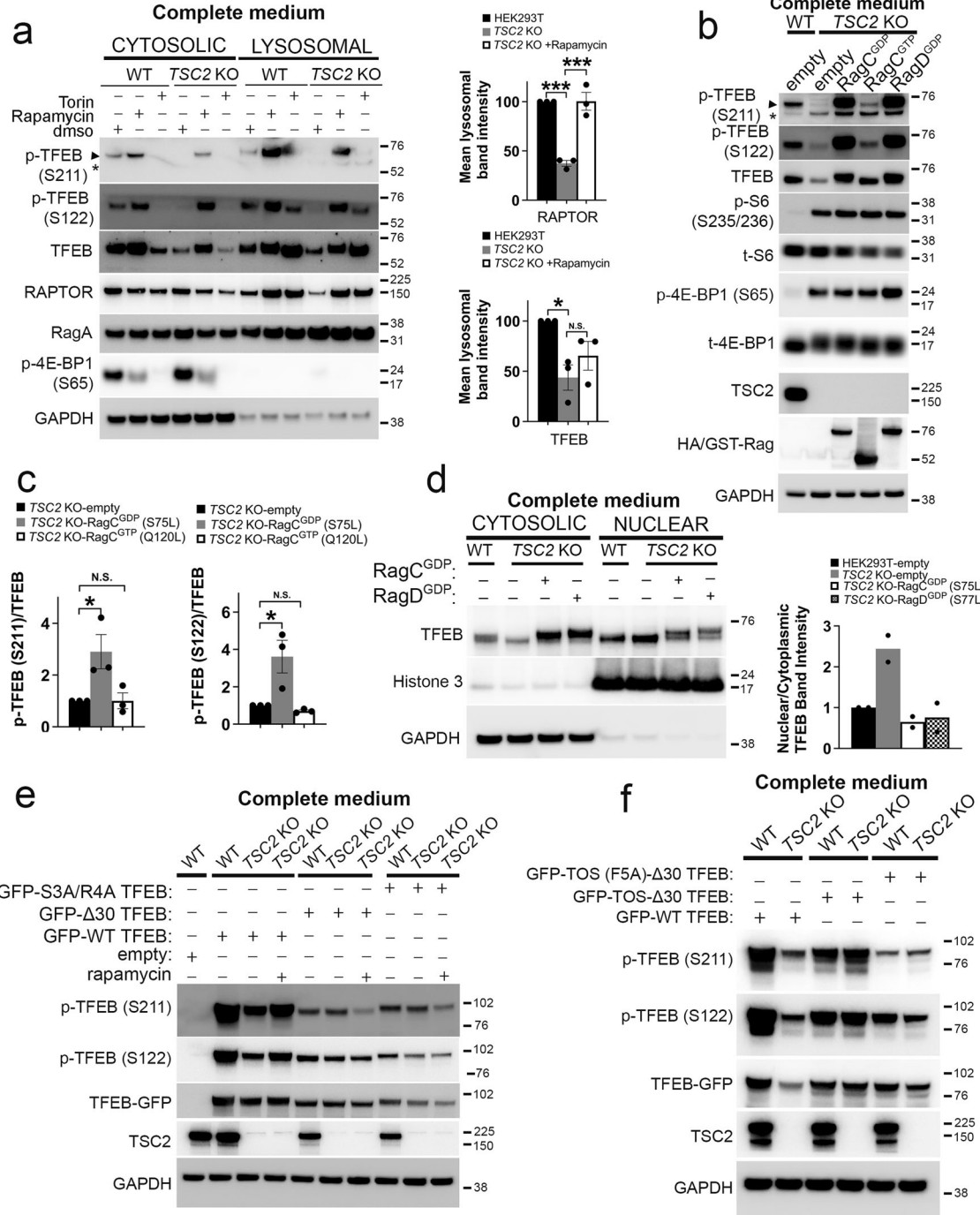

**Fig. 6 | Active RagC/D heterodimers rescue TFEB lysosomal localization, phosphorylation and cytoplasmic retention in cells with TSC2 loss. a** WT and *TSC2* KO cells in complete growth media were treated with vehicle control, rapamycin (200 nM) or torin (1 μM) for 2 h prior to lysosomal fractionation and immunoblotting of cytosolic and lysosomal fractions. Densitometry quantification of normalized lysosomal Raptor and TFEB band intensity is shown in right panels. *n* = 3 independent biological replicates. Graphs are presented as mean values ± SEM. Statistical analyses were performed using one-way ANOVA with Dunnett's test for multiple comparisons. *$p$ < 0.05, ***$p$ < 0.001. **b** TSC2 KO cells were transiently transfected with control vector or HA-tagged, active RagC^GDP (S75L), inactive RagC^GTP (Q120L) or active RagD^GDP (S77L) for 24 h prior to lysis and immunoblotting (also see Fig. S8b). **c** Densitometry quantification of representative immunoblots from experiments in **b**. *n* = 3 independent biological replicates. Graphs are presented as mean values ± SEM. Statistical analyses were performed using one-way ANOVA with Dunnett's test for multiple comparisons. *$p$ < 0.05. **d** TSC2 KO cells

were transiently transfected with control vector or HA-tagged, active RagC^GDP or active RagD^GDP for 24 h prior to nuclear fractionation and immunoblotting of cytosolic and nuclear fractions. Densitometry quantification of representative immunoblots are shown on the right. **e** WT and TSC2 KO cells were transiently transfected with empty vector, WT-TFEB-GFP, a TFEB mutant lacking the first 30 amino acids (Δ30-TFEB-GFP) or a TFEB point mutant within the first 30 residues of TFEB (TFEB-S3A, R4A-GFP) for 48 h, followed by immunoblotting of cell lysates. TSC2 KO cells were treated with vehicle or rapamycin (200 nM; 2 h), prior to lysis. **f** WT and TSC2 KO cells were transiently transfected with WT-TFEB-GFP, TOS-Δ30-TFEB-GFP (a mutant where the first 30 amino acids of TFEB have been substituted with the first 30 amino acids of S6K containing the TOS motif), or TOS (F5A)- Δ30-TFEB-GFP (a variant of TOS- Δ30-TFEB-GFP in which a key phenylalanine residue (F5) of the TOS motif has been mutagenized to alanine) for 48 h, followed by cell lysis and immunoblotting. Source data are provided as a Source Data file.

### Destabilization of the lysosomal folliculin complex upstream of Rag GTPases partially restores nutrient-induced TFEB phosphorylation in TSC2 KO cells

The RagC/D GTPase activating protein (GAP) folliculin (FLCN)[46,50] is essential for MiT/TFE phosphorylation due to their dependence on Rag recruitment. Consequently, *FLCN* loss promotes TFEB/TFE3 nuclear localization[13,14,46,51]. On nutrient starvation, recruitment of FLCN and its binding partner, FNIP2 (the FLCN:FNIP2 complex), to the inactive Rag heterodimer at the lysosome results in formation of a stable lysosomal folliculin complex (LFC)[51]. The RagC GAP activity of FLCN is inhibited within the LFC, and disruption of the LFC upon amino-acid exposure has been shown to be critical for mTORC1 activation of the MiT/TFE factors[51].

We first examined whether destabilization of the LFC was sufficient to restore TFEB phosphorylation in *TSC2* KO cells. We leveraged two recently described *FLCN* mutations with opposing effects on RagC GAP activity: (a) The FLCN[F118D] mutation at the FLCN-RagA interface, which fails to assemble the LFC and exhibits uninhibited GAP activity, and (b) The FLCN[R164A] mutation that assembles into a normal LFC but lacks GAP activity[51,52]. We generated stable cell lines expressing these mutants in WT and *TSC2* KO cells depleted for endogenous *FLCN* with shRNA. Compared to WT cells expressing scrambled control shRNA, *FLCN* depletion in WT cells (first four lanes of Fig. S9a), was sufficient to de-phosphorylate TFEB, as expected, and similar results were observed with *FNIP2* depletion (Fig. S9b) indicating that both components of the dimer are required for TFEB phosphorylation. Stable expression of FLCN[F118D] in *FLCN*-depleted, *TSC2* KO cells also partially restored TFEB phosphorylation, while FLCN[R164A] had no effect (lanes 7–10 of Fig. S9a). Thus, TFEB phosphorylation in *TSC2* KO cells remains sensitive to destabilization of the LFC in response to diverse stimuli, which suggests that that endogenous FLCN/FNIP2 and RagC activity can be at least partially reactivated in these cells.

### Restoration of FLCN:FNIP2 lysosomal localization by mTORC1 inhibition or co-expression of FLCN and FNIP2 fully rescues TFEB phosphorylation in cells with TSC2 loss

In response to amino-acid starvation, the FLCN:FNIP2 complex gets recruited to the lysosome, where it interacts with the Rag-Ragulator complex, and this lysosomal relocalization is an essential prerequisite for its role as a RagC/D GAP upon nutrient restimulation leading to subsequent mTORC1 activation. We examined the lysosomal recruitment of FLCN and FNIP2 in *TSC2* KO cells via specific interaction of this complex with components of Rag-Ragulator in immunoprecipitation experiments. In WT cells transiently expressing Lamtor1-GFP, the binding of FLCN and FNIP2 was increased in Lamtor1 immune-precipitates, following starvation as expected (Fig. 7a, b). However, this interaction was significantly compromised in *TSC2* KO cells and was completely rescued upon mTORC1 inhibition with rapamycin or torin (Fig. 7a–c, Fig. S9c, d). These findings suggested that the failure of FLCN:FNIP2 to localize to the lysosome and activate the Rag heterodimer in *TSC2* KO cells could underlie TFEB hypophosphorylation in this context.

To test this hypothesis, we examined whether lysosomal relocalization or activation of FLCN:FNIP2 could rescue TFEB phosphorylation in *TSC2* KO cells. It has been noted in previous studies that concurrent overexpression of exogenous FLCN and FNIP2 leads to constitutive lysosomal localization of FLCN, independent of nutrient conditions[46,50,53–55]. Similarly, FNIP proteins are also critical for recruitment of HA-tagged FLCN expressed from an endogenous locus[46]. Corresponding to the observed defect in FLCN:FNIP2 recruitment to the lysosome in *TSC2* KO cells, transient co-expression of WT FLCN and FNIP2 (but not either one alone), fully rescued TFEB phosphorylation in parental *TSC2* KO cells (lanes 3–5 of Fig. 7d). In *FLCN*-depleted *TSC2* KO cells, co-transfection of FNIP2 with stably-expressed WT FLCN or activated FLCN[F118D] (lanes 8–9 or 12–13 of Fig. 7d, lanes

9–13 of Fig. S9a) led to a rescue of p-TFEB, while a similar rescue was not seen with inactive FLCN[R164A] as a negative control. To confirm the functional significance of these results, we examined the subcellular localization of TFEB/TFE3 in cells expressing FLCN mutants. *TSC2* KO cells depleted for *FLCN* with shRNA, and stably expressing FLCN[F118D], partially suppressed nuclear localization of TFEB/TFE3, and this effect was further enhanced by transient co-expression of FNIP2 in these cells (Fig. 7e, f).

Interestingly, total expression of FNIP2 was decreased in vivo in renal tumors in *Tsc2±* and *Pax8 Cre*; *Tsc2*[fl/wt] mice by immunohistochemistry (Fig. 7g). We also examined tumor xenograft growth in *FLCN*-depleted, *TSC2* KO cells stably expressing FLCN[F118D] or a combination of FLCN[F118D]/FNIP2. Tumor growth was modestly, although non-significantly, suppressed in cells stably expressing FLCN[F118D]/FNIP2 consistent with the more attenuated rescue of TFEB phosphorylation in vivo (Fig. S9e, f) compared to what was observed in vitro, potentially due to issues with long-term expression of these constructs. Taken together, these data suggest that the FLCN:FNIP2 complex fails to be recruited to the lysosomal membrane during starvation in *TSC2* KO cells, in an mTORC1-dependent manner. Exogenous expression of FLCN and FNIP2 restores these proteins to the lysosomal membrane and rescues TFEB phosphorylation and cytosolic localization in cells with *TSC2* loss.

## Discussion

Nutrient-mediated substrate phosphorylation by mTORC1 is canonically associated with enhanced anabolism and reduced catabolism[1]. However, substrate selectivity by mTORC1 and consequently, the regulation of metabolism associated with constitutive mTORC1 activity in the setting of *TSC2* loss is more complex than previously appreciated. Notably, a previous study from 2011 was the first to show that lysosomal V-ATPase genes are upregulated with *Tsc2* loss in MEFs[15]. In support of this and our own studies on murine epidermal *Tsc1* loss[16], and findings from other groups[15,17–20], we find that MiT/TFE transcriptional factors *TFEB* and *TFE3* are paradoxically and constitutively activated in three independent murine and human renal cell line models of *TSC1/2* inactivation with a resulting increase in lysosomal biogenesis. We validated these findings in a mouse model of spontaneous *Tsc2* loss, as well as in two additional mouse models of conditional *Tsc1/2* deletion-induced renal tumorigenesis. Thus, these findings appear to be highly conserved between MiT/TFE family members and across tissue types and species.

TFEB phosphorylation at the canonical mTORC1 site Ser211 results in binding to 14-3-3 chaperone proteins, cytosolic retention and inactivation[6–9]. De-phosphorylation at another mTORC1 site, S122, is simultaneously essential for TFEB nuclear localization following mTORC1 inhibition[12]. We found that TFEB phosphorylation at S211 and S122 was paradoxically suppressed in murine and human cell line and xenograft models of *TSC2* loss. Unexpectedly, mTORC1 inhibition with rapamycin actually promoted TFEB phosphorylation in *TSC2* KO cells and xenografts, restored TFEB cytosolic localization and potently inhibited *TSC2* KO tumor xenograft growth. A recent landmark study demonstrated that phosphorylation of TFEB unlike other mTORC1 substrates (S6K, 4EBP1, ULK1; all of which are hyperactivated with *TSC1/2* loss), exclusively relies on the amino-acid-dependent recruitment by activated RagC/D[13]. However, in contrast to this study where TFEB phosphorylation was shown to be insensitive to short-term *TSC2* knockdown, we found that TFEB was hypophosphorylated in cells with genomic deletion of *TSC1/2* in an amino-acid-dependent manner. Consistent with this, expression of active GDP bound RagC or RagD mutants completely restored TFEB phosphorylation and cytosolic localization in *TSC2* KO cells. Mechanistically, rapamycin also promoted the interaction of TFEB with Rag GTPases and Raptor, restored lysosomal localization of mTORC1 and TFEB, and consequently increased TFEB phosphorylation in *TSC2* KO cells, in a manner

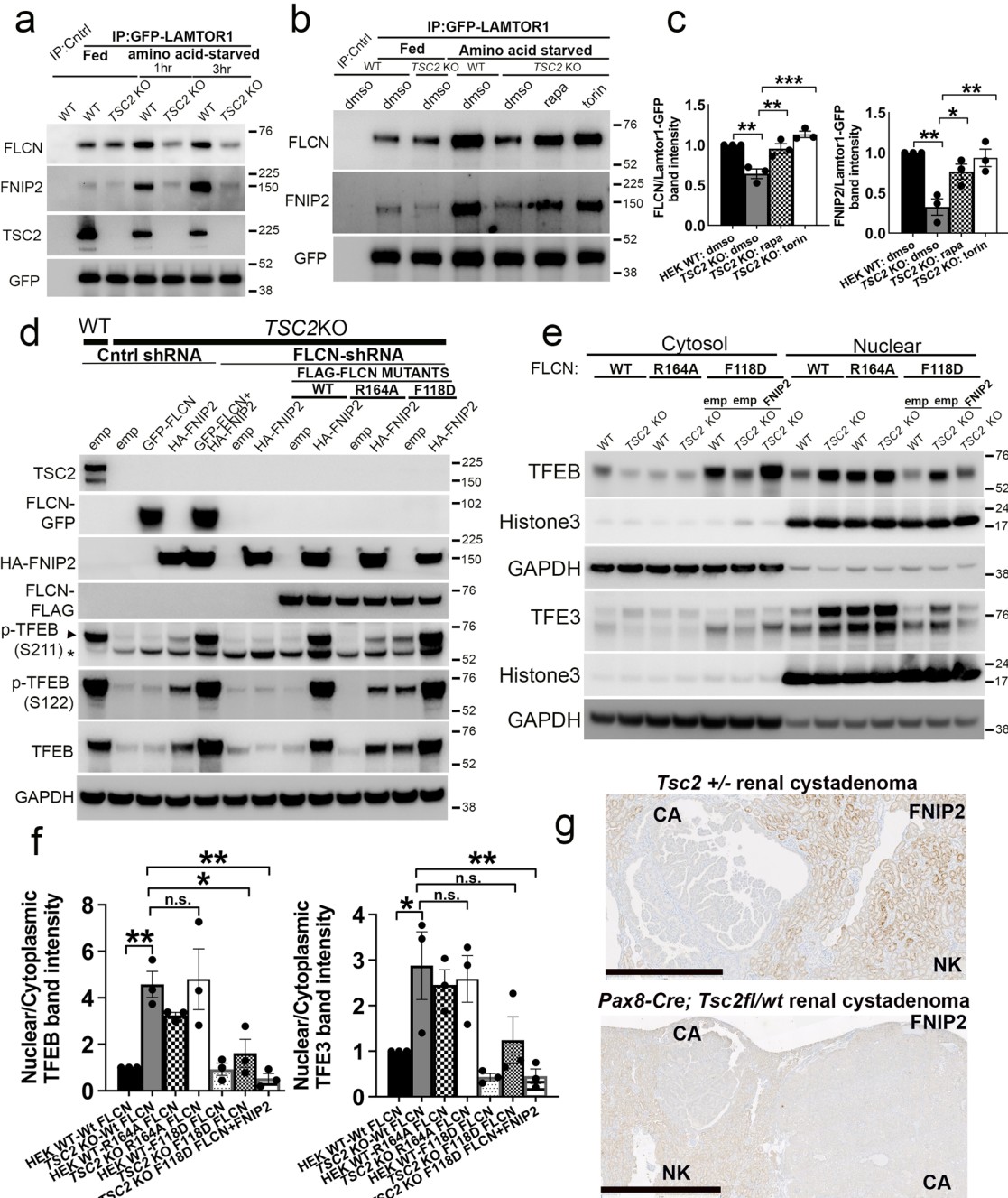

**Fig. 7 | Restoration of FLCN:FNIP2 lysosomal localization by mTORC1 inhibition or co-expression of FLCN and FNIP2 fully rescues TFEB phosphorylation in cells with TSC2 loss. a** WT and *TSC2* KO cells were transiently transfected with Lamtor1-GFP for 24 h, and maintained in complete growth media (fed) or amino-acid-starved for 1 or 3 h, followed by immunoprecipitation (IP) with a binding control or a GFP-Trap coupled to magnetic agarose beads and immunoblotting. **b** WT and *TSC2* KO cells were transiently transfected with Lamtor1-GFP for 24 h. The indicated cells were then treated with dmso (vehicle), rapamycin (200 nM), or torin (1 μM) for 1 hr in complete growth media and amino-acid-starved for 1 h, prior to lysis and immunoprecipitation (IP) with either a binding control or a GFP-Trap coupled to magnetic agarose beads and immunoblotting (also see Fig. S9c). **c** Densitometry quantification of normalized FLCN or FNIP2 band intensities from immunoblot experiments in amino-acid-starved samples shown in **b**. *n* = 3 independent biological replicates. **d** WT and *TSC2* KO control shRNA-expressing cells were transiently transfected with empty vector, GFP-FLCN, HA-FNIP2 or both GFP-FLCN and HA-FNIP2 for 24 h, as indicated (lanes 1–5). In the same experiment, *FLCN*-depleted

*TSC2* KO cells, stably expressing FLAG-tagged, WT FLCN (lanes 8–9), FLCN[R164A] (lanes 10–11), or FLCN[F118D] (lanes 12–13) mutants were transiently transfected with either empty vector or HA-FNIP2 for 24 h followed by lysis and immunoblotting (also see Fig. S9a). **e** TFEB and TFE3 expression in nuclear-fraction immunoblots of *FLCN*-depleted, WT and *TSC2* KO cells stably expressing FLAG-tagged, WT FLCN, FLCN[R164A], or FLCN[F118D] mutants. In the same experiment, the FLCN[F118D] cells were transiently transfected with empty vector or HA-FNIP2, prior to fractionation, lysis and immunoblotting. **f** Densitometry quantification of representative immunoblot experiments from **e**. *n* = 3 independent biological replicates. **g** Representative IHC for FNIP2 in murine renal cystadenomas (CA), compared to surrounding normal kidney (NK) in *Tsc2* ± mice (top panel; scale bar = 1 mm) and *Pax8 Cre; Tsc2*[fl/wt] mice (bottom panel; scale bar = 2 mm). All graphs are presented as mean values ± SEM. Statistical analyses were performed using one-way ANOVA with Dunnett's test for multiple comparisons. *p < 0.05, **p < 0.01, ***p < 0.001. Source data are provided as a Source Data file.

that required the N-terminal, Rag-binding domain of TFEB. Cumulatively, these data suggest that amino-acid-dependent RagC/D activation and TFEB phosphorylation are constrained via an auto-regulatory, rapamycin-sensitive mTORC1 negative feedback mechanism in cells with *TSC2* loss.

The tumor suppressor FLCN and its binding partner FNIP2 comprise the primary GAP complex activating RagC/D and are accordingly crucial for the phosphorylation of TFEB/TFE3 by mTORC1[46,50]. The lysosomal recruitment of the FLCN:FNIP2 dimer, binding to the Rag-Ragulator proteins and formation of the lysosomal folliculin complex (LFC) following amino-acid starvation are critical initial steps in RagC activation. Subsequently, nutrient-induced disruption of the LFC is an essential prerequisite for FLCN-RagC and mTORC1 activation[51]. Significantly, the lysosomal localization of FLCN:FNIP2 in response to nutrient starvation was severely compromised in *TSC2* KO cells. As a result, forced lysosomal localization of the FLCN:FNIP2 complex by concurrent overexpression of both proteins, or rescue of endogenous FLCN:FNIP2 lysosomal localization by mTORC1 inhibition, completely restored TFEB phosphorylation and cytosolic localization in these cells. Notably, destabilization of the LFC induced in response to FLCN[F118D] also partially rescued TFEB phosphorylation, presumably by activating the low endogenous levels of lysosomal FLCN/FNIP2 in *TSC2* KO cells.

Collectively, these experiments support our current model: in the context of *TSC2* loss, constitutive mTORC1 signaling inhibits lysosomal recruitment of FLCN:FNIP2 with nutrient starvation, thus preventing activation of RagC and TFEB recruitment to the lysosome for phosphorylation by mTORC1 once nutrients are replete. Rapamycin is sufficient to restore FLCN:FNIP2 to the lysosome, and does not inhibit mTORC1 activity towards TFEB (due to its incomplete inhibition of mTORC1), thus leading paradoxically to increased TFEB phosphorylation and inactivation. Short-term in vitro torin has similar effects on FLCN:FNIP2 localization but simultaneously potently inhibits mTORC1 activity towards TFEB, thus failing to robustly rescue TFEB phosphorylation in the context of *TSC2* KO. Precisely how mTORC1 activity inhibits lysosomal recruitment of FLCN:FNIP2 remains to be elucidated. Notably, phosphorylation of both FLCN and FNIP2 has been implicated as an mTORC1-mediated negative feedback pathway in prior studies of yeast and mammalian cells[56,57], and these phosphorylation sites are rapamycin-sensitive[57,58]. Future studies will examine whether post-translational modifications of FLCN:FNIP2 occur in the context of *TSC2* loss and may modulate recruitment of the FLCN:FNIP2 dimer to the lysosome.

Beyond expanding our current models of MiT/TFE regulation, our findings have potential clinical relevance since mTORC1 inhibition and MiT/TFE inactivation may show synergy for treatment of some tumor types with mTORC1 activation. MiT/TFE hyperactivity drives kidney cyst and tumor formation in mice and humans[13,59,60] as well as in pancreatic cancer and melanoma[26,41]. Given the weak effects of torin on *TSC2* KO xenograft growth and TFEB phosphorylation, it is likely that the more potent effects of rapamycin are attributable to a combination of MiT/TFE inactivation and the expected downstream effects of mTORC1 inhibition. Thus, mTOR kinase inhibitors such as torin, which inhibit mTORC1 but fail to robustly inactivate TFEB, may be less preferable compared to rapalogs for treatment of cells with *TSC2* inactivation. Although isolated CRISPR-Cas9-mediated inactivation of *TFEB* or *TFE3* did not affect growth of *TSC2* KO xenografts, it is notable that combined inactivation of both factors (similar to that achieved with rapamycin) had a significant effect, consistent with potent suppression of lysosomal genes by RNA-seq. This is largely consistent with previous studies which have suggested that functional redundancy between TFEB and TFE3 is significant[38,39], though this may not be the case in all contexts or for all readouts. For example, *Tfeb* KO alone was sufficient to attenuate renal tumorigenesis

driven by *FLCN* loss[13]. Along the same lines, loss of *TFEB* or *TFE3* individually was sufficient to reciprocally decrease phosphorylation of mTORC1 substrates S6K and 4E-BP1 in cells with *TSC2* loss. This was likely partly due to decreased transcription of the CLEAR gene target *RRAGD*, as has previously been documented in cells with intact *TSC1/2*[37], and further suggests that targeting the MiT/TFE pathway might potentially synergize with mTOR inhibitor therapy in this setting. Ultimately, future studies will reveal whether *Tfeb* and/or *Tfe3* deletion is sufficient to reduce renal tumorigenesis in mice with spontaneous or inducible loss of *Tsc2*, and establish whether MiT/TFE targeting could be therapeutically useful in the setting of human tuberous sclerosis.

## Methods

### Cell culture
Human embryonic kidney HEK293T cells with or without somatic genomic deletion (KO) of *TSC1, TSC2*, and *TSC1/2* via CRISPR-Cas9 genome editing were a kind gift of TS Alliance and Dr. Nellist[30]. TRI102 cells derived from a *TSC2*-null human AML and TRI103[44] cells derived from TRI102 cells stably transfected with wild-type *TSC2* (pcDNA3.1 TSC2-zeo) were obtained from ATCC (Manassas, VI) (Catalog numbers: PTA-7368 and PTA-7369). TTJ cells, a *Tsc2*-null cell line derived from *Tsc2* ± C57BL/6 mice, stably transfected with a control vector (TTJ-parental) or wild-type *Tsc2* (TTJ-*Tsc2*) were a kind gift of Dr. Vera Krymskaya[35]. Cell culture conditions are described in supplemental methods.

### Plasmids and lentiviral transfections
Plasmids obtained from Addgene are described in supplemental methods. pLJM1 FLCN (WT) FLAG, pLJM1 FLCN (F118D) FLAG and pLJM1 FLCN (R164A) FLAG were a kind gift of Dr. Roberto Zoncu[51]. GFP-TOS-Δ30TFEB and GFP-F5A-Δ30TFEB were a kind gift of Dr. Andrea Ballabio[13]. Cells were transiently transfected using Lipofectamine 3000 reagent (L3000008, Thermo Fisher Scientific) according to the transfection guidelines. Lentiviral infection and generation of clones stably expressing scrambled, *FLCN*, *FNIP2*, or *RRAGD* shRNAs and WT/ mutant FLCN vectors was performed using the PEI method as previously described[51].

### CRISPR-Cas9 genome editing for TFEB and TFE3
We designed single-guide RNA (sgRNA) for 3 target sequences in the human *TFE3* gene (GGCGATTCAACATTAACGACAGG, GCGACGCT-CAACTTTGGAGAGGG, TCGCCTGCGACGCTCAACTTTGG) and 4 target sequences in the human *TFEB* gene (CAACCCTATGCGTGACGCCATGG, GCGGTAGCAGTGAGTCGTCCAGG, TGCCTAGCGAAGAGGGCCCAGGG and GAGTACCTGTCCGAGACCTA) and cloned these into the lenti-CRISPR v2 vector (Addgene #52961, Watertown, MA, USA). Lentivirus was produced as previously described[61] and *TSC2* KO cells were infected for 48 h and selected with puromycin (1 μg/mL) for 10 days, and monoclonal cell colonies established. Immunoblotting was used for confirmation of *TFEB* and *TFE3* KO.

### Antibodies and reagents
Antibodies and reagents used in this study are described in supplemental methods.

### Animal studies
Animal protocols were approved by the JHU Animal Care and Use Committee. The following strains were used: (1) The *Tsc2* ± A/J mice, heterozygous for a deletion in exons 1–2 were a kind gift from David Kwiatkowski (Harvard University, Boston, USA) and have been previously characterized[34], (2) Mice carrying loxP sites flanking exon 17 and 18 of *Tsc1*(Stock Number 005680, *Tsc1*[tm1Djk]/J) (The Jackson Laboratory). (3) Mice carrying loxP sites flanking exon 2, 3, and 4 of *Tsc2*(Stock Number 027458, *Tsc2*[tm1.1Mjgk]/J) (The

Jackson Laboratory). (4) Mice bearing a tamoxifen-inducible Cre recombinase driven by the endogenous mouse *Gt(ROSA)26Sor* promoter (Stock Number: 004847, R26CreER) (The Jackson Laboratory). (5) Mice heterozygous for the *Pax8* cre recombinase knockin gene (Stock Number: 028196, Pax8^cre) (The Jackson Laboratory). Conditional deletion of *Tsc1* was obtained by tamoxifen treatment of *Rosa(ER)-Cre; Tsc1^fl/fl* mice as previously described[62]. Renal tubular-specific deletion of *Tsc2* was obtained by crossing <u>heterozygously</u>-expressing *Pax8-cre* mice with *Tsc2*^fl/wt mice to generate *Pax8-Cre; Tsc2 ^fl/wt* mice. Genotyping primers and xenograft studies are described in supplemental methods.

### Laser-capture microdissection
Laser-capture microdissection was performed as described in supplemental methods.

### Histology and immunostaining
Histology and immunostaining was performed as described in supplemental methods.

### Cell and tissue lysates, immunoblotting, and immunoprecipitation
Cell and tissue lysis protocols, immunoblotting, immunoprecipitation, nuclear fractionation assays are described in supplemental methods. Uncropped and unprocessed western blots for all the figures have been provided as a separate PDF within the Source Data file.

### Lysosomal fractionation assays
Lysosomal fractionation assays were carried out as previously described[63] and in supplemental methods.

### Luciferase reporter assays
Luciferase reporter assays are described in supplemental methods.

### RNA isolation and quantitative real-time RT-PCR
RNA isolation and quantitative real-time RT-PCR and primers used are detailed in supplemental methods.

### Immunocytochemistry
Immunofluorescence protocols and quantification techniques are described in supplemental methods.

### RNA-sequencing and data analysis
RNA-sequencing of triplicate cell line and xenograft replicates was performed at Novogene and carried out as previously described[64]. Raw RNA-seq counts were FPKM-normalized and used for plotting while DESeq2 was employed to determine differentially expressed genes.

### Gene set enrichment analysis, GSEA
Gene Set Enrichment Analysis, GSEA: (http://www.broad.mit.edu/gsea/) was used to evaluate whether lysosomal gene expression was differentially regulated in cell lines and xenografts. Raw RNA-seq counts from our experiments were used as input in GSEA together with the following seven literature-curated gene sets for lysosomal activity: (1) A set of 435 lysosomal genes from The Human Lysosome Gene Database (hLGDB) was used[40]. Separately, hLGDB sets for *TFEB* binding sites ($n = 70$ genes) and CLEAR motifs ($n = 69$ genes) were also employed for GSEA analyses, (2) We then used CLEAR and *TFEB* gene sets from ref. 31, which were composed of 95 and 299 genes, respectively[31], (3) A set of 135 genes from ref. 41 was used to investigate whether our expression data was associated with MITF/TFE-related autophagy pathways[41]. Lastly, *MITF*-related gene sets were obtained from ref. 42. The seven sets were compared to the universe of all the arrays, collapsed to genes, and provided Normalized Enrichment Score

(NES) and q-values for each individual set compared with our expression data. Negative NES indicated negatively enriched pathways in our comparison group vs. control. Q-value cutoffs were set to 0.1.

### Cathepsin D activity assays
Cathepsin D enzyme activity assays were performed as described in supplemental methods.

### Cell viability assays
Cell viability assays were performed as described in supplemental methods.

### Confluence assays
Confluence assays were performed as described in supplemental methods.

### Statistics and reproducibility
For image analysis, RNA and protein quantification, luciferase assays and xenograft studies, statistical significance was determined using the unpaired, two-tailed Student's *t*-test when comparing two experimental groups, or with one-way ANOVA with Dunnett's or Bonferroni's correction when comparing 3 or more experimental groups. Mean values were performed in GraphPad Prism (version 8.2.1). *p*-values of <0.05 were considered statistically significant. All experiments were repeated at least three times (independent biological replicates) with similar results. Additionally, all experiments with the *TFEB, TFE3*, and dual *TFEB/TFE3* KO cells were performed using multiple clones, and multiple orthogonal techniques were utilized to ensure rigor. For example: (a) TFEB/TFE3 nuclear localization was confirmed by immunofluorescence (cells), IHC (mouse renal tumors and xenografts) and immunoblotting of nuclear-cytoplasmic fractions, (b) increased TFEB/TFE3 transcriptional activity was confirmed by qRT-PCR, immunoblotting of lysosomal proteins, RNASeq and 4XCLEAR promoter activity assay.

### Reporting summary
Further information on research design is available in the Nature Portfolio Reporting Summary linked to this article.

## Data availability
All data generated and analyzed during the current study are included in this published article and its supplementary information files, or are deposited in GEO. All unique materials generated (such as the *TFEB/TFE3* KO CRISPR cell lines) during this study are available from the corresponding author upon request. A reporting summary for this article is available as a Supplementary information file. Literature-curated lysosomal gene sets from the following studies were used for Gene Set Enrichment Analysis: (1) The Human Lysosome Gene Database (hLGDB) Brozzi et al.[40] (https://pubmed.ncbi.nlm.nih.gov/23584836/, (2) Sardiello et al.[31] (https://pubmed.ncbi.nlm.nih.gov/19556463/) (GSE16267), (3) Perera et al.[41] (https://pubmed.ncbi.nlm.nih.gov/26168401/) (GSE62077), and (4) Hoek et al.[42] (https://pubmed.ncbi.nlm.nih.gov/19067971/). The RNA-seq data from this study are deposited into NCBI's Gene Expression Omnibus (GEO) database with the accession code GSE216545. Source data are provided with this paper.

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

## Acknowledgements
This research was supported in part by the CDMRP TSCRP grant W81XWH-22-1-0264 (T.L.L.), CDMRP KCRP grant W81XWH-20-1-0843 (T.L.L.), CDMRP TSCRP grant W81XWH-19-1-0781 (K.A.), R01CA200858 (T.L.L.), and the NCI Cancer Center Support Grant 5P30CA006973-52. Additional funding was provided by Joey's Wings Foundation and Dahan Translocation Carcinoma fund. We thank the TSC Alliance and Dr. Nellist for providing us the Human embryonic kidney HEK293T cells with or without somatic genomic deletion (KO) of *TSC1*, *TSC2*, and *TSC1/2* via CRISPR-Cas9 genome editing, the TSC Alliance, Van Andel Research Institute and Dr. David Kwiatkowski for providing us *Tsc2* ± A/J mice, Dr Vera Krymskaya for providing the TTJ- parental and TTJ-*Tsc2* cell lines, Dr. Roberto Zoncu for providing the pLJM1 FLCN (WT) FLAG, pLJM1 FLCN (F118D) FLAG and pLJM1 FLCN (R164A) FLAG plasmids, and Dr. Andrea Ballabio for providing the GFP-TOS-Δ30TFEB and GFP-F5A-Δ30TFEB plasmids.

## Author contributions
T.L.L. and K.A. conceived the study. K.A. and T.L.L. drafted the manuscript. K.A., J.W., A.A.M., E.S., T.V., C.R.V., K.F., L.O., S.M., H.B.L., D.C.S., B.L., and P.A. completed the data collection and analysis. All authors critically reviewed the manuscript and agreed to submit for publication.

## Competing interests
T.L.L. has received research support from Roche/ Ventana, Myriad Genetics, DeepBio and AIRA Matrix for other studies. The remaining authors declare no competing interests.
