## [Peer Review File · Nature Communications]

An mTORC1-mediated negative feedback loop constrains amino acid-induced FLCN-Rag activation in renal cells with TSC2 lossREVIEWER COMMENTS

Reviewer #1 (Remarks to the Author):

In this study Asrani et al. explore the mechanism underlying paradoxical TFEB nuclear localization and activation in TSC-deficient cells (and the similarly paradoxical effect of rapamycin) and assess the role of TFEB/TFE3 in TSC-dependent tumorigenesis. Confirming the work from others, they show that TSC2 loss results in TFEB nuclear localization and activation and that this effect is reversed by rapamycin. They evaluate a variety of tumors with TSC inactivation (EVT, LOT and ESC) where they report TFEB nuclear localization. They show that combined TFEB/TFE3 depletion reduces confluency and tumor growth in xenografts. They show that TSC loss reduces baseline S122/211 phosphorylation, which is induced by rapamycin. They also show that expression of active RAGC and RAGD induces S122/211 phosphorylation and cytoplasmic TFEB retention in TSC-deficient cells. They also show that deletion/mutation of N-terminal 30 amino acids, which are required for TFEB recruitment to the lysosome, is sufficient to downregulate S122/211 phosphorylation, and that replacing this sequence for a TOS motif restores TFEB phosphorylation in TSC-deficient cells. They show that FLCN is required for baseline S122/211 phosphorylation and that expression of SLC38A9N is sufficient to restore S122/211 phosphorylation in TSC-deficient cells.

The paper addresses an important question, it is well organized and clearly written. However, much of the information presented has already been reported by others, including Brugarolas (much of the data in Figs 1 and 2) as well as Ballabio and Henske (peppered throughout).

Main points

While the authors propose a compelling model for how TFEB is deregulated by TSC/rapamycin, they don't provide evidence of a direct link and the same results may be achieved through modulation of parallel pathways.

Understanding, at the molecular, perhaps structural level, how rapamycin induces the changes in TSC2 deficient cells would significantly raise the impact of the manuscript.

It would be interesting to see what happens when TFEB/TFE3 are disrupted (individually and combined) in TSC-dependent GEMM models of tumorigenesis.

The authors state “We found that these TSC1/2/mTOR alteration-associated renal tumor subtypes frequently expressed nuclear TFEB or TFE3 at levels comparable to TFEB- or TFE3-translocation RCC.” The authors should substantiate this claim by providing, at a minimum, a Table illustrating the number of tumors examined (EVT, LOT and ESC) compared to tRCC and quantitating the results.

The authors claim that Tsc2 loss results in increased lysosome activity, but this is not directly shown.

The authors show that TFE3/TFEB depletion dampens mTORC1 activity in TSC2 KO cells, but the mechanism is unclear.

It is not clear whether the control for the HEK293T TSC2KO includes an irrelevant sgRNA. This applies to other scenarios such as the generation of FLCN knockdowns. If appropriate controls have been used, this should be included in the main text. If not, key experiments should be repeated with appropriate controls.

Other points

It may be best to move previously reported findings to the supplement. References to work from others should also be made more explicit.

TFE3 (and to a certain extent TFEB) loss appears to actually induce growth in xenografts. This is not explained.

The authors should check for the expected loss of the remaining Tsc2 allele in heterozygous tumor models.

The impact of TFEB/TFE3 inactivation on cell proliferation of TSC-deficient cells should be assessed by comparison to TSC-competent controls.

Assessments of cell proliferation should go beyond “percentage confluency” which may be affected by other factors including cell size.

The effect on confluency seems to be quite different between the B3-1 and B4-3-3 TFEB sgRNA targeted clones and this remains to be explained.

Decreased 14-3-3 binding of TFE3 in TSC2 KO cells should be shown with endogenous proteins. It would be good to perform the experiment with rapa and torin as well.

IP experiments of lysosome bound proteins are confounded by the possibility that organelle membranes may be recovered confounding the interpretation.

The authors show that expression of active RAGC and RAGD induces S122/211 phosphorylation in TSC-deficient cells. Similar overexpression experiments should be shown in WT cells for comparison.

Fig 5E first panel WT vs KO bar seems misplaced on second sample.

Some statements are not fully accurate: “while previous studies with rapamycin have analyzed phosphorylation in the context of exogenous TFEB and by band-shift analyses as a marker of TFEB phosphorylation, we used specific and validated p-TFEB antibodies, examining both endogenous and overexpressed TFEB.”

Reviewer #2 (Remarks to the Author):

Asrani et al. present an elegant work of cell signaling in which they found that amino acid-stimulated phosphorylation of TFEB is impaired in cells with constitutive mTORC1 activity due to TSC2 loss, resulting in increased MiT/TFE activity. Surprisingly, TFEB phosphorylation in TSC2-null cells is rescued by rapamycin, constitutive activation of RagC/D, or destabilization of the upstream lysosomal folliculin complex induced by the SLC38A9. Thus, the authors suggest a novel negative feedback loop that constrains amino acid-induced mTORC1 activity in the setting of TSC2 loss. However, the authors did not solve a piece of this pathway. Who (mTORC1?) and how SLC38A9 is constrained and or modulates FLCN to reduce Rag activation? In addition, what is the kinetics of the process? What is first in the tumorigenic process upon TSC loss, mTORC1 hyperactivation, or TFEB nuclear translocation?

Also, these findings may represent an opportunity to identify novel therapeutics to treat tumors driven by the mTORC1/TFEB/TFE3 pathway. However, the authors stopped in the partially solved mechanism and did not test the effects of rapamycin and SLC38A9N/FLCNF118D mutants on TSC2 null-driven tumors. The manuscript is potentially interesting for Nature Communications. However, all these questions need to be addressed before considering the publication in this journal.

Some other comments:

- 1) Perhaps mTORC1 hyperactivation regulates the activity of SLC38A9, FLCN, or both by phosphorylation. Are these proteins phosphorylated by mTOR (since the mechanism is rapamycin-sensitive)? What residues are involved?
- 2) The authors must test their hypothesis reverting tumor growth (xenograft) by expressing SLC38A9N and FLCNF118D mutants rescuing TFEB/TFE3 hypo phosphorylation and nuclear translocation.
- 3) Similar experiments of point 2 must be done using rapamycin. A comparison with torin1 must be done in parallel. It could be a great proof of principle for future therapy.
- 4) IFs of TFEB/TFE3 subcellular localizations are missing in SLC38A and FLCN mutant overexpressing cells.
- 5) FLCN KO cells and tissues present an aberrant nuclear localization of TFEB. Also, the authors claim that “rapamycin-induced phosphorylation of TFEB is dependent on FLCN, similar to phosphorylation of TFEB in WT cells”. This claim is based on figure 7A, where the authors said that stable depletion of FLCN partially inhibited rapamycin-induced TFEB phosphorylation in TSC2 KO cells. However, the total levels of TFEB are highly reduced on this blot, making the ratio of P-TFEB/Total TFEB very similar to the effects of rapamycin alone in TSC ko cells. It may suggest that rapamycin and FLCN are not in the same pathway. How do the authors explain this? Perhaps, the same experiment but looking at the subcellular localization of TFEB/TFE3 may help to clarify this question. Also, using the same experimental strategy, what are the effects of TSC depletion in FLCN KO cells?
- 6) The message of the paper may improve a lot with a scheme of the pathway identified
- 7) There are very recent high-impact articles, even in the big brother Nature, around the pathway studied here. It would be great for general readers to compare your most significant findings and claims with the observations of these articles. Some examples: a) There are contradictory observations regarding the effects of FLCN loss and TSC loss on TFEB nuclear localization. b) Depletion of TFEB in a kidney-specific mouse model of BHD syndrome fully rescued the disease phenotype and associated lethality and normalized mTORC1 activity, whereas in the cancer models studied in this manuscript the double depletion of TFEB and TFE3 is required to reduce growth.

Minor:

- Some English editing is needed. The abstract and introduction are difficult to understand for general readers. There are many difficult to read sentences
- What was the rationale to use rapamycin? did you perform a survey of several drugs, or its selection was by chance? An explanation would be useful in the manuscript.

Reviewer #3 (Remarks to the Author):

In their manuscript titled “An mTORC1-mediated negative feedback loop constrains amino acid induced SLC38A9 activity in tuberous sclerosis complex”, Asrani and colleagues aim to provide a deeper understanding of how transcription factor EB (TFEB) driven lysosomal biogenesis can be induced as a compensatory process upon constitutive activation of mTORC1. The core idea of the manuscript is interesting, and the authors show that due to deficient recruitment of TFEB and TFE3 to the lysosomes, they escape their inhibiting phosphorylation, which allow them to translocate to the nucleus and activate their target genes. Those data, although not without merit, do not generate a convincingly novel concept as neither, the precise mechanism behind the defective recruitment, nor the link to amino acid metabolism or tuberous sclerosis has sufficiently been addressed. To improve the quality and increase the novelty of their finding, the authors should perform more experiments to better understand how TFEB can escape the inhibitory phosphorylation that is catalyzed upon short term activation of mTOR.

Major comments

1. The title is misleading, as the main focus of the paper is not to link the TFEB-driven negative feedback of constitutive mTORC1 activation to tuberous sclerosis. The authors might consider to either tone down the title, or to perform further experiments with a clearer focus on disease progression and/or amelioration.
2. The authors show that stable loss of TCS2 promotes TFEB mediated transcription of lysosomal genes. However, although they have obtained TCS1 KO, as well as TCS1/2 double KO cells, they do not confirm their gene expression and protein abundance findings (Fig1) in those cell lines. It would be an important control to provide those panels, as the authors have also done for the nuclear localization IF and WB.
3. Why is the TFEB-driven transcriptional increase not observed in normal kidney cells although other non-cancerous cells (e.g., primary keratinocytes) manifest that too? Is it restricted to rapidly proliferating cells?
4. Because of the difference in inhibitory potential between rapamycin and torin1 (especially with respect to the kinase function), it would be interesting, to address the localization of TFEB, as well as the transcriptional activation of its targets following torin1 treatment. Those results might also help to better characterize the precise mechanism of how constitutive mTORC1 inhibition leads to an increase in TFEB phosphorylation. Have the authors tried similar experiments?
5. From the WB in Suppl. Fig2D and E it seems more as if total TFEB decreases in TCS2 KO cells. In absolute nuclear amounts, there seems to be no difference between WT and KO cells. Would not similar levels of nuclear TFEB result in similar degrees of gene expression activation independently of the amount of cytoplasmic TFEB?

6. The KO experiments of TFEB and TFE3 are intriguing as it partially seems as if they (over-)compensate for one another (e.g., in the tumor growth experiment) while in other readouts (e.g., the WB in Fig4A) they seem to not always be able to compensate. A better characterization of the single vs the double knockouts would help to clarify those partially conflicting results.

7. It would be of great value for the manuscript if the authors would try to address how loss of TSC could influence the interaction between TFEB and the Rag GTPases and how rapamycin would contribute to the reversal of the phenotype.

8. In line with point 7, while the authors nicely show how forced or artificial tethering of TFEB to the lysosomes results in its phosphorylation, whereas inhibition of TFEB going to the lysosomes results in its hypo-phosphorylation, it remains unclear, how the TFEB is kept away from the lysosomes upon lack of TSC. It might be insightful to try to pull down TFEB in cells proficient or deficient for TSC and potentially identify different interaction partners that could help to elucidate the molecular mechanism.

9. Similarly, the involvement of FLCN, LFC and SLC38A9 would require further experiments. Also here, identification of the interactome of FLCN and SLC38A9 following knockout of TSC might help to better understand their involvement. Alternatively, the involvement of SLC38A9 could be demonstrated by using specific antagonists and subsequently addressing mTORC1 activity in general as well as with respect to TFEB.

Minor comments

1. Revise sentence “We demonstrate the Rag-mediated TFEB phosphorylation is in response to amino acids is..”; page 5

2. The DAPI staining in Fig. 2B seems overexposed

3. The parenthesis “(hereafter referred to as TSC2 KO and WT..)” should be moved to the first results paragraph, as the respective abbreviations have already been used there.

Reviewer(s)' Comments to Author:

Reviewer: 1 (Remarks to the Author):

In this study Asrani et al. explore the mechanism underlying paradoxical TFEB nuclear localization and activation in TSC-deficient cells (and the similarly paradoxical effect of rapamycin) and assess the role of TFEB/TFE3 in TSC-dependent tumorigenesis. Confirming the work from others, they show that TSC2 loss results in TFEB nuclear localization and activation and that this effect is reversed by rapamycin. They evaluate a variety of tumors with TSC inactivation (EVT, LOT and ESC) where they report TFEB nuclear localization. They show that combined TFEB/TFE3 depletion reduces confluency and tumor growth in xenografts. They show that TSC loss reduces baseline S122/211 phosphorylation, which is induced by rapamycin. They also show that expression of active RAGC and RAGD induces S122/211 phosphorylation and cytoplasmic TFEB retention in TSC-deficient cells. They also show that deletion/mutation of N-terminal 30 amino acids, which are required for TFEB recruitment to the lysosome, is sufficient to downregulate S122/211 phosphorylation, and that replacing this sequence for a TOS motif restores TFEB phosphorylation in TSC-deficient cells. They show that FLCN is required for baseline S122/211 phosphorylation and that expression of SLC38A9N is sufficient to restore S122/211 phosphorylation in TSC-deficient cells.

The paper addresses an important question, it is well organized and clearly written. However, much of the information presented has already been reported by others, including Brugarolas (much of the data in Figs 1 and 2) as well as Ballabio and Henske (peppered throughout).

We thank the reviewer for this thoughtful appraisal of our work. We agree entirely that our work builds on important prior publications from the Brugarolas, Ballabio and Henske groups, and we have cited these studies throughout, but we want to emphasize that there are significant ways in which our work diverges from theirs, and adds substantially to our current understanding of mTORC1 regulation of MiT/TFE transcription factors.

- 1) The landmark Brugarolas study (PMID: 21804531), from 2011, was the first to show that V-ATPase gene upregulation (a key subset of lysosomal genes) is seen with *Tsc2* loss in MEFs and this is reversed by rapamycin, requires TFEB, and is associated with phosphatase-sensitive gel shifts in TFEB suggestive of phosphorylation changes. In our current study, we add to this work by demonstrating that:
 - a. TFEB is *paradoxically hypophosphorylated* in *TSC2* KO cells (shown by epitope-specific antibodies), leading to TFEB nuclear translocation and upregulation of MiT/TFE-regulated gene sets.
 - b. We further validate these findings at the protein level in cell lines and at the gene expression and protein level in *in vivo* models including *TSC2* KO xenografts and *Tsc1,2* KO renal cystadenomas. These *in vivo* models comprise much of Figures 1-3.
- 2) In Ballabio's critical 2020 study (PMID:32612235), TFEB is shown to be recruited to the lysosome for phosphorylation by mTORC1 specifically in response to amino acid stimulation, via activated RagC/D downstream of folliculin activation, with *in vivo* validation in murine kidney tumors. However, in this report, *siRNA-mediated depletion of TSC2 did*

not affect phosphorylation or subcellular localization of TFEB. In stark contrast, we report that:

- a) *TSC2 loss not only does not increase TFEB phosphorylation, but paradoxically decreases it, leading to TFEB activation.*
 - b) *Amino acid stimulation fails to induce TFEB lysosomal recruitment and phosphorylation in the context of TSC2 loss.*
 - c) This hypophosphorylation of TFEB in the context of TSC2 loss is *due to failure of RagC/D activation downstream of amino acid stimulation* and reversible by transfection with active RagC/D.
- 3) Finally, while we were writing our manuscript in 2021, Lisa Henske's group published a study (PMID:34253722) confirming the Brugarolas report that there is increased lysosomal gene expression (and increased lysosomes) in *Tsc2*^{-/-} MEFs and demonstrating that this can be reproduced in *Tsc2*^{+/-} mouse renal tumors. In HeLa cells, they show that exogenous GFP-TFEB is hypophosphorylated with knockdown of *TSC1* or *TSC2* via siRNA leading to nuclear accumulation, and show that active RagC transfection is sufficient to reverse nuclear accumulation of TFEB-GFP in HeLa cells. We feel we have added significantly to this study by demonstrating that:
- a) *Endogenous TFEB is hypophosphorylated in TSC2 KO cells, and accumulates in the nucleus to activate gene transcription at promoters with CLEAR motifs.*
 - b) *TFEB activation occurs in TSC2 KO cells because RagC/D are not activated, specifically in response to amino acid stimulation and thus fail to recruit TFEB to the lysosome for phosphorylation by mTORC1.*
 - c) *The FLCN:FNIP2 complex, a key upstream GAP and activator of RagC/D, fails to be recruited to the lysosome in response to amino acid starvation in TSC2 KO cells. TFEB hypophosphorylation and nuclear accumulation can be rescued in this context by increasing FLCN levels at the lysosome (via simultaneous FLCN/FNIP2 transfection which leads to constitutive lysosomal localization, or by mTORC1 inhibition) or by activating FLCN mutations, or upstream SLC28A9 activation of folliculin.*
 - d) *Hypophosphorylation and nuclear accumulation of TFEB are reversible by rapamycin in TSC2 KO cells and xenografts, confirming that mTORC1 activity regulates this paradoxical finding and distinguishing effects of rapamycin (a partial mTORC1 inhibitor) from that of torin.*
 - e) *Endogenous TFE3 is also activated in TSC2 KO cells, and TFEB and TFE3 can partially compensate for one another in this context.*

Main points:

1) While the authors propose a compelling model for how TFEB is deregulated by TSC/rapamycin, they don't provide evidence of a direct link and the same results may be achieved through modulation of parallel pathways. Understanding, at the molecular, perhaps structural level, how rapamycin induces the changes in TSC2 deficient cells would significantly raise the impact of the manuscript.

This is an excellent point and one that we have worked hard to address with additional mechanistic data. As previously described (PMID: 24081491, 24095279, 29848618, 31672913), multiple lines of evidence support a role for FLCN and its binding partners FNIP2 (the FLCN:FNIP2 complex) in coordinating cellular responses to amino acid availability via Rag heterodimer activation, and corresponding mTORC1 activation at the lysosome. In response to amino acid starvation, the FLCN:FNIP2 complex is first recruited to the lysosome, where it interacts with the

Rag-Ragulator complex, and this re-localization is an essential prerequisite for its role as a RagC/D GAP upon nutrient re-stimulation and subsequent mTORC1 activation.

We have now provided additional mechanistic details underlying the regulation of TFEB hypo-phosphorylation in *TSC2* KO cells, and its reversal by mTORC1 inhibition. We show that the lysosomal recruitment of both FLCN and FNIP2 upon amino acid starvation is significantly compromised in HEK293T *TSC2* KO cells (new Figure 7A), and is completely reversible upon mTORC1 inhibition with rapamycin or torin (new Figure 7B-C, new Supplementary Figure S10 C, D). Consistent with these *in vitro* findings, expression of FNIP2 was decreased *in vivo* in renal tumors in *Tsc2* +/- mice by immunohistochemistry (new Figure 7G). Importantly, in WT cells, FLCN depletion (first four lanes of new Supplementary Figure S10A) or FNIP2 depletion (new Supplementary Figure S10B) by shRNA was sufficient to suppress TFEB phosphorylation, indicating that both components of the dimer are required for TFEB phosphorylation. Taken together, these data suggested the hypothesis that failure of FLCN:FNIP2 to localize to the lysosome and activate the Rag heterodimer in *TSC2* KO cells could explain TFEB hypophosphorylation in this context.

To test this, we examined whether lysosomal localization or activation of FLCN:FNIP2 could rescue TFEB phosphorylation in *TSC2* KO cells. It has been noted in previous studies using overexpressed FLCN and FNIP2 that co-expression of both proteins (but not either one alone) leads to constitutive lysosomal localization of FLCN (PMID: 24081491, 18663353, 24095279, 27113757, 28039480). Corresponding to the observed defect in FLCN:FNIP2 recruitment to the lysosome in *TSC2* KO cells, transient expression of WT FNIP2 alone partially rescued, and co-expression of WT FLCN and FNIP2 fully rescued TFEB phosphorylation in *TSC2* KO cells (lanes 4 and 5 of new Figure 7D). In *TSC2* KO cells depleted for FLCN with shRNA, similar synergistic effects with FNIP2 on TFEB phosphorylation were also seen for WT FLCN or activated FLCN^{F118D} (lanes 8-9 or 12-13 of new Figure 7D, lanes 9-13 of Supplementary Figure S10A), while a similar rescue was not seen with inactive FLCN^{R164A} as a negative control. To confirm the functional significance of these results, we also performed experiments to determine the subcellular localization of TFEB/TFE3 in cells stably expressing FLCN mutants. *TSC2* KO cells depleted for FLCN with shRNA, and stably expressing FLCN^{F118D} partially suppressed nuclear localization of TFEB/TFE3, and this effect was further enhanced by transient co-expression of FNIP2 in these cells (new Figure 7E, F). These data cumulatively suggest that expression and/or lysosomal localization of the FLCN/FNIP2 complex is perturbed with *TSC2* loss, and its rescue by FLCN/FNIP2 overexpression or mTORC1 inhibition directly resulted in the rescue of TFEB hypo-phosphorylation in these cells.

We have added this information to the Results (lines 402-459) and Discussion (515-543) of the manuscript.

2) It would be interesting to see what happens when TFEB/TFE3 are disrupted (individually and combined) in TSC-dependent GEMM models of tumorigenesis

We agree that it would be of great interest to determine the role of *TFEB* and/or *TFE3* in driving renal tumorigenesis in the context of genetically engineered mouse models driven by *Tsc1/2* loss, similar to what has been recently described in the context of renal tumors driven by *FLCN* loss (PMID: 32612235). Unfortunately, based on new data described below, we estimate that these experiments will likely require years to complete due to our finding of substantial functional redundancy for TFEB/TFE3 in the context of *TSC2* loss.

Previous studies have suggested that functional redundancy between TFEB and TFE3 is significant (PMID: 27298091, 27171064), though this was notably not observed in the recent *FLCN* study where *Tfeb* KO alone was sufficient to attenuate renal tumorigenesis (PMID: 32612235). However, in the context of *TSC2* loss specifically, we have performed growth assays and xenograft experiments (**new Figure 4B, C, Supplementary Figure S4**), and found that single gene inactivation of *TFEB* or *TFE3* had no effect on growth of *TSC2* KO xenografts, while double KO of *TFEB* and *TFE3* had a significant effect. In order to further characterize potential functional redundancy of TFEB and TFE3 in the *TSC2* KO background, we have now performed RNAseq on tumor xenograft specimens isolated from WT, *TSC2* KO, *TFEB* KO, *TFE3* KO and double *TFEB/TFE3* KO mice (**new Figure 4D**). Multiple, *TFEB*- and/or *TFE3*-regulated lysosomal gene sets were positively enriched by GSEA in *TSC2* KO compared to WT xenografts, validating our initial *in vitro* findings. While *TFEB/TFE3* DKO xenografts on the *TSC2* KO background showed significant negative enrichment of all lysosomal gene sets compared to *TSC2* KO xenografts, findings with individual knockout of TFEB or TFE3 were weaker. These results are consistent with our observed effects on tumor growth and confirm redundancy among MiT/TFE family members in this context.

Given these results, to test effects in GEM models, we would need to generate kidney-specific, *Tfeb/Tfe3* DKO mice in the background of *Tsc2* deletion. Since *Tfeb* *-/-* mice are not viable, this would need to be done via a cre-lox system for at least *Tfeb* (eg, generating *Pax8-cre; Tfeb*^{fl/fl}; *Tfe3*^{-/-}; *Tsc2*^{+/-} mice). To do this using the spontaneous *Tsc2* *+/-* A/J strain would require that all strains be first backcrossed onto the A/J background, given reported variability in renal tumor development by mouse strain for *Tsc2* (PMID: 20146790). While these backcrossing issues could potentially be circumvented in the novel and highly penetrant *Pax8-cre; Tsc2*^{fl/wt} mice reported in our current study (which are of the same C57BL6 background as the commercially available *Tfeb*^{fllox/fllox} and *Tfe3*^{-/-} mice), this would nevertheless require creating a quadruple transgenic animal (*Pax8-cre; Tfeb*^{fl/fl}; *Tsc2*^{fl/wt}; *Tfe3*^{-/-}) which we estimate require more than two additional years to generate, age and characterize for renal tumors. Given these timelines, it would not be feasible to include these experiments in the current manuscript.

We have added this information to the Results (lines 240-251) and Discussion (553-569) section of the manuscript.

3) The authors state “We found that these TSC1/2/mTOR alteration-associated renal tumor subtypes frequently expressed nuclear TFEB or TFE3 at levels comparable to TFEB- or TFE3-translocation RCC.” The authors should substantiate this claim by providing, at a minimum, a Table illustrating the number of tumors examined (EVT, LOT and ESC) compared to tRCC and quantitating the results.

We agree that we only provided anecdotal and qualitative evidence of TFEB/TFE3 expression in human renal tumors associated with TSC1/2/mTOR alterations, largely because we do not have access to significant numbers of these relatively rare tumors. In addition, variability in TFEB and TFE3 immunostaining -- likely due to pre-clinical variables such as fixation or older sample age - - has been an omnipresent technical challenge previously discussed in many publications (see PMID: 30851332), and has precluded quantification of this staining in other human studies. In fact, in our clinical practice at Johns Hopkins, we rely on *TFEB/TFE3* FISH rather than IHC to screen for translocation renal cell carcinomas due to these pervasive issues. Since we lack adequate case numbers and the optimal tools for detection and rigorous quantification of nuclear TFE3 and TFEB in human tumors, we have removed anecdotal discussion of the human tumors from our current study, relying instead on robust results in 3 independent *in vivo* models as well

as xenografts, where the antibody performance is much better due to highly controlled fixation conditions and new, rather than archival, samples.

Notably, we have recently published another study (PMID: 35072947) examining GPNMB expression in RCC. This is a direct TFEB/TFE3 target gene and, in contrast to TFEB/TFE3, GPNMB is a very robust marker by immunostaining in human tissue. In our other publication, we quantitatively demonstrate that GPNMB has equally high and specific expression in translocation-associated RCC as well as EVT, LOT and ESC tumors and provides an excellent biomarker of both tumor types since it is not expressed in CCRCC, papRCC or ChRCC. This provides proof of principal that TFEB/TFE3 activity is increased in EVT, LOT and ESC tumors.

We have added this information to the Introduction on lines 107-109.

4) The authors claim that Tsc2 loss results in increased lysosome activity, but this is not directly shown.

We previously provided indirect evidence of increased lysosomal activity in *TSC2* KO cells, as highlighted below:

1) Increased proteolytic processing and conversion of Cathepsin B to a mature, 27 kDa form in HEK293T *TSC1* KO, *TSC2* KO and *TSC1/2* KO cells compared to WT controls (**new Figure 1D**) and in lysosomal fractions of *Tsc2* null TTJ cells (**new Figure 1J**), indicative of a functional increase in lysosomal proteolysis.

2) Immunofluorescence of endogenous LC3 showing increased presence of puncta in *TSC2* KO cells, indicative of increased autophagosomes (**new Supplementary Figure S1B**)

3) Increased LC3-II accumulation (arrow) with hydroxychloroquine (HCQ) and chloroquine (CQ) (**new Supplementary Figure S1C**) indicative of a higher autophagic flux (a measure of autophagic degradation activity) at baseline, in *TSC2* KO cells.

At the request of the reviewer, we have now provided more direct evidence of increased lysosomal activity in *TSC2* KO cells. Cathepsin D enzyme activity was determined using a Cathepsin D activity assay kit (ab65302, Abcam), a fluorescence-based assay that utilizes the preferred cathepsin-D substrate sequence GKPILFFRLK(Dnp)-D-R-NH₂ labeled with MCA. Cathepsin D activity was significantly increased in *TSC2* KO cells compared to WT controls (**new Supplementary Figure S1D**). We have now added this to the Results section of the manuscript (lines 159-161).

5) The authors show that TFE3/TFEB depletion dampens mTORC1 activity in TSC2 KO cells, but the mechanism is unclear.

This is an interesting point. Notably, a prior study (PMID: 28619945) has demonstrated that MiT/TFE3 activation and CLEAR-mediated gene transcription may lead to increased RagD expression (since Rag D contains a CLEAR element in its promoter) and thus contribute to increased mTORC1 signaling. Though this had not been previously shown for *TSC2* KO cells, we hypothesized that CLEAR activation downstream of *TSC2* loss may lead to increased RagD transcription and contribute to increased mTORC1 signaling in these cells. If this is the case, then TFE3/TFEB depletion in the setting of *TSC2* loss might in turn decrease RagD transcription and dampen mTORC1 signaling.

To begin to address this hypothesis, we first examined *TSC2* KO cells and xenografts and observed that, consistent with our finding of CLEAR activation in these cells, *RRAGD* mRNA and RagD protein levels were increased in *TSC2* KO and *TSC1, 2* KO cells and xenografts, compared to their WT counterparts (**new Figure 1 C, D** and **Supplementary Figure S6A, B**). In *TSC2* KO cells, we show that *RAGD* shRNA partially suppresses mTORC1 substrate phosphorylation by immunoblotting (**new Supplementary Figure S6C**), and overexpression of active RagD further boosted 4EBP1 phosphorylation in *TSC2* KO cells (**new Figure 6B**). These data suggest that there is at least some residual activity of RagD in *TSC2* KO cells *vis a vis* non-MiT/TFE mTORC1 substrates such as 4EBP1 and S6K. *In vitro*, we find that *RAGD* transcription can be suppressed in *TSC2* KO cells with *TFE3* or *TFEB* KO or *TFEB/TFE3* KO (**new Supplementary Figure S6D**), paralleling the diminished mTORC1 signaling we observed in *TFE3* or *TFEB* KO or *TFEB/TFE3* KO cells *in vitro* (**new Figure 4A**). These results indicate that similar to what has been previously reported in WT cells, increased RagD transcription also potentially contributes to mTORC1 signaling in *TSC2* KO cells.

We have added these points to the Results (lines 252-271; lines 330-332) and Discussion (560-565) section of the manuscript.

6) It is not clear whether the control for the HEK293T TSC2KO includes an irrelevant sgRNA. This applies to other scenarios such as the generation of FLCN knockdowns. If appropriate controls have been used, this should be included in the main text. If not, key experiments should be repeated with appropriate controls.

We can confirm that we have used appropriate controls for all cell lines that we have generated in the current study. In all experiments involving *FLCN* knockdown via shRNA (Figure 7A from the previous version of the manuscript, as well as **new Figure 7D** and **new Supplementary Figure S10A** in this revised version), WT and *TSC2* KO cells stably expressing a scrambled shRNA were used as controls. Similarly, cells stably expressing scrambled shRNA were used as controls for *TSC2* KO cells expressing RagD shRNA (**new Supplementary Figure S6C**) and WT cells expressing *FNIP2* shRNA (**new Supplementary Figure S10B**). Cells expressing non-targeting gRNA were used as controls for experiments with genomic deletion of *TFEB*, *TFE3* and *TFEB/3* (**new Figure 4**). HEK293T *TSC2* KO cells were generated, characterized and reported previously by another group (PMID: 27406250).

We have now added this to the Results section (lines 231; lines 261-262; lines 404-405).

Other points:

1) It may be best to move previously reported findings to the supplement. References to work from others should also be made more explicit.

Thank you for this suggestion. We have moved some overlapping findings to the supplement. However, due to the distinct nature of the cell lines used (HEK293T *TSC1*, *TSC2* and *TSC1/2* KO CRISPR cells compared to *TSC2*-null HeLa/MEF cells used by other groups), certain overlap in initial findings is unavoidable. Discussion of recent seminal papers in the field (as highlighted in point #1, above) have also been given more prominence.

These references are now in the Introduction and Discussion sections of the manuscript as follows: (lines 89-93; lines 97-102; lines 481-482; lines 559-560)

2) TFE3 (and to a certain extent TFEB) loss appears to actually induce growth in xenografts. This is not explained.

While this trend is intriguing, it is not statistically significant, thus we have refrained from commenting about this. It is conceivable that this is due to feedback up-regulation of the other MiT/TFE family proteins, however given that it is not statistically significant, we cannot exclude that it could be due to chance variability in xenograft growth.

3) The authors should check for the expected loss of the remaining Tsc2 allele in heterozygous tumor models.

In a recent publication from our lab, we confirmed Tsc2 protein loss in renal cystadenomas and tumors in *Tsc2* +/- A/J mice using a novel immunohistochemistry assay (PMID: 35072947, Figure 2A), consistent with bi-allelic inactivation of *Tsc2*. In response to this comment, we have now examined expression of Tsc2 by immunohistochemistry in *PAX8-Cre; Tsc2*^{fl/wt} mice (**new Figure 3C**). By 18 months of age, full-blown cystadenomas and cystadenocarcinomas are evident, with accompanying Tsc2 protein loss (confirming bi-allelic *Tsc2* inactivation).

We have now added this information to the Results section of the manuscript (lines 164; lines 220-222).

4) The impact of TFEB/TFE3 inactivation on cell proliferation of TSC-deficient cells should be assessed by comparison to TSC-competent controls. Assessments of cell proliferation should go beyond “percentage confluency” which may be affected by other factors including cell size.

We have now assessed the impact of *TFEB/TFE3* inactivation on *TSC2* KO cell proliferation using a more conventional, colorimetric method for assaying viable cells- CellTiter 96® Aqueous One Solution Cell Proliferation Assay (MTS) (**new Supplementary Figure S4B**). The MTS assay protocol is based on the reduction of the MTS tetrazolium compound by viable mammalian cells (and cells from other species) to generate a colored formazan dye that is soluble in cell culture media. This conversion is thought to be carried out by NAD(P)H-dependent dehydrogenase enzymes in metabolically active cells. The formazan dye is quantified by measuring the absorbance at 490-500 nm. Additionally, HEK293T WT cells were also simultaneously assayed, as controls. The results of the MTS viability assays were mostly comparable to the confluence assays.

We have now added these findings to the Results section of the manuscript (lines 236-238).

5) The effect on confluency seems to be quite different between the B3-1 and B4-3-3 TFEB sgRNA targeted clones and this remains to be explained.

We agree that the effects of *TFEB* deletion in clone B4-3-3 were more profound than clone B3-1 in *in vitro* confluency assays (**new Supplementary Figure S4A**), and we saw similar results in MTS assays (**new Supplementary Figure S4B**), though neither clone showed significant reduction in viability compared to the *TSC2* KO control clone. We also observed greater

suppression of tumor xenograft growth in clone B4-3-3, however, these results were not significant (**new Figure 4B**). While it is possible that this variation could be the result of clonal variation or divergence, it is interesting that clone B4-3-3 also displayed a more significant suppression of *RRAGD* mRNA levels compared to control *TSC2* KO cells (**new Supplementary Figure S6D**), induction of which has been previously associated with hyper-proliferation and cancer growth (PMID: 28619945). Further studies will be required to assess the contribution of *RRAGD* in driving tumor cell proliferation in the context of *TFEB/TFE3* deletion in cells with *TSC2* loss.

6) Decreased 14-3-3 binding of TFE3 in TSC2 KO cells should be shown with endogenous proteins. It would be good to perform the experiment with rapa and torin as well.

We attempted to examine the binding of 14-3-3 proteins in immunoprecipitations of endogenous TFEB from WT and *TSC2* KO cells using a Cell Signaling antibody (#37785), and were able to detect TFEB in both non-denaturing (lysed in Cell Lysis Buffer #9803, Cell Signaling) as well as denaturing cell lysates (using RIPA buffer for more complete cell lysis) (**Response Figure # 1A**). However, we were unable to detect prominent and/or single bands for 14-3-3 proteins, in contrast to immunoprecipitations of exogenous TFEB-GFP, where we detected single and prominent band in IP reactions and whole cell lysates (**new Figure 5E**). We then examined the binding of endogenous TFEB to exogenous, tagged 14-3-3 in reverse Co-IPs. WT and *TSC2* KO cells were transiently transfected with a HA-tagged 14-3-3 gamma expression vector for 24 hrs, followed by lysis and immunoprecipitation using an anti-HA or control antibody (**Response Figure # 1B**). Consistent with our previous results in **new Figure 5E (left panel)**, binding of endogenous TFEB was decreased in exogenous, tagged 14-3-3 immune-precipitates in *TSC2* KO cells compared to WT cells, and was rescued by rapamycin, but not torin. We have now added this figure to the manuscript (**new Figure 5E (right panel); new Supplementary Figure S7F (lower panel)**).

Response Figure 1: (A) Immunoprecipitation of endogenous TFEB using an anti-TFEB antibody (#37785, Cell Signaling), or isotype control IgG, in WT and *TSC2* KO cell lysates and immunoblotting. *TSC2* KO cells were treated with rapamycin (200 nM; 2 hrs) or torin (1 μ m; 2 hrs), prior to lysis. **(B)** WT and *TSC2* KO cells were transiently transfected with a HA-tagged 14-3-3 gamma expression vector for 24 hrs, followed by lysis and immunoprecipitation using an anti-HA antibody or control IgG. *TSC2* KO cells were treated with rapamycin (200 nM; 2 hrs) or torin (1 μ m; 2 hrs), prior to lysis.

7) IP experiments of lysosome bound proteins are confounded by the possibility that organelle membranes may be recovered confounding the interpretation.

We agree that pull-down of endosome-lysosome membranes in IP experiments for lysosomal-bound proteins may be a confounding factor if the purpose of the experiment is to establish protein-protein interactions. However, we are largely using lysosomal protein IP experiments to examine lysosomal recruitment of other proteins, thus presence of endo-lysosomal membranes is actually desirable. This is the case for GFP-Lamtor1 pull-down experiments examining for FLCN/FNIP2 recruitment to the lysosome (**new Figure 7A-C**) as well as HA-RagB pull-down experiments (**new Supplementary Figure S10D**).

8) The authors show that expression of active RAGC and RAGD induces S122/211 phosphorylation in TSC-deficient cells. Similar overexpression experiments should be shown in WT cells for comparison.

We have now repeated the experiments with overexpression of active RagC and RagD in both WT and *TSC2* KO cells and added these findings in **new Supplementary Figure S8B** and to the Results section of the manuscript (lines 329-330)

9) Fig 5E first panel WT vs KO bar seems misplaced on the second sample.

In **new Figure 5E** (first panel), the first 2 lanes are both WT cells: the first lane corresponds to immunoprecipitation using a binding control IgG magnetic agarose (as a control), while the second lane corresponds to immunoprecipitation using GFP Trap magnetic agarose.

10) Some statements are not fully accurate: “while previous studies with rapamycin have analyzed phosphorylation in the context of exogenous TFEB and by band-shift analyses as a marker of TFEB phosphorylation, we used specific and validated p-TFEB antibodies, examining both endogenous and overexpressed TFEB.”

We apologize for this inaccurate statement. This statement has now been removed.

Reviewer: 2 (Remarks to the Author):

Asrani et al. present an elegant work of cell signaling in which they found that amino acid-stimulated phosphorylation of TFEB is impaired in cells with constitutive mTORC1 activity due to *TSC2* loss, resulting in increased MiT/TFE activity. Surprisingly, TFEB phosphorylation in *TSC2*-null cells is rescued by rapamycin, constitutive activation of RagC/D, or destabilization of the upstream lysosomal folliculin complex induced by the *SLC38A9*. Thus, the authors suggest a novel negative feedback loop that constrains amino acid-induced mTORC1 activity in the setting of *TSC2* loss.

We thank the reviewer for these positive comments regarding our work.

1) However, the authors did not solve a piece of this pathway. Who (mTORC1?) and how *SLC38A9* is constrained and or modulates FLCN to reduce Rag activation?

Thank you for this comment. We have now provided additional mechanistic details underlying the regulation of TFEB hypo-phosphorylation in *TSC2* KO cells, and its reversal by mTORC1 inhibition. As previously described (PMID: 24081491, 24095279, 29848618, 31672913), multiple lines of evidence support a role for FLCN and its binding partners FNIP (the FLCN: FNIP2

complex) in coordinating cellular responses to amino acid availability via Rag heterodimer activation, and corresponding mTORC1 activation at the lysosome. In response to amino acid starvation, the FLCN: FNIP2 complex is first recruited to the lysosome, where it interacts with the Rag-Ragulator complex, and this lysosomal re-localization is an essential prerequisite for its role as a RagC/D GAP upon nutrient re-stimulation leading to subsequent mTORC1 activation.

We now demonstrate that the lysosomal recruitment of both FLCN and FNIP2 upon amino acid starvation is significantly compromised in HEK293T *TSC2* KO cells (**new Figure 7A**), and this is completely reversed upon mTORC1 inhibition with rapamycin or torin (**new Figure 7B-C**, **new Supplementary Figure S10 C,D**). Consistent with these *in vitro* findings, expression of FNIP2 was decreased *in vivo* in renal tumors in *Tsc2* +/- mice by immunohistochemistry (**new Figure 7G**). Importantly, FLCN depletion (first four lanes of **new Supplementary Figure S10A**) or FNIP2 depletion (**new Supplementary Figure S10B**) by shRNA is sufficient to suppress TFEB phosphorylation in WT cells, indicating that both components of the dimer are required for TFEB inactivation. Taken together, these data suggested the hypothesis that failure of FLCN: FNIP2 to localize to the lysosome and activate the Rag heterodimer in *TSC2* KO cells could underlie TFEB hypophosphorylation in this context.

To test this, we examined whether lysosomal re-localization or activation of FLCN: FNIP2 could rescue TFEB phosphorylation in *TSC2* KO cells. It has been noted in previous studies that concurrent over-expression of FLCN and FNIP2 leads to constitutive lysosomal localization of FLCN, independent of nutrient conditions (PMID: 24081491,18663353, 24095279, 27113757, 28039480). Corresponding to the observed defect in FLCN: FNIP2 recruitment to the lysosome in *TSC2* KO cells, co-expression of WT FLCN and FNIP2 fully rescued TFEB phosphorylation in *TSC2* KO cells (lanes 4 and 5 of **new Figure 7D**). To test the effects of activating or inactivating FLCN mutants, we did similar experiments in *TSC2* KO cells depleted for endogenous FLCN with shRNA. Co-transfection of FNIP2 with WT FLCN or activated FLCN^{F118D} (lanes 8-9 or 12-13 of **new Figure 7D**, lanes 9-13 of **Supplementary Figure S10A**) led to a rescue of p-TFEB, while a similar rescue was not seen with inactive FLCN^{R164A} as a negative control. To confirm the functional significance of these results, we examined the subcellular localization of TFEB/TFE3 in cells expressing FLCN mutants. *TSC2* KO cells depleted for FLCN with shRNA, and stably expressing FLCN^{F118D} partially suppressed nuclear localization of TFEB/TFE3, and this effect was further enhanced by transient co-expression of FNIP2 in these cells (**new Figure 7E, F**).

Taken together, these data suggest that the FLCN:FNIP2 complex fails to be recruited to the lysosomal membrane during starvation in *TSC2* KO cells, and this is mTORC1-dependent and reversible by concurrent overexpression of FLCN and FNIP2 which restores these proteins to the lysosomal membrane. Consistent with this model, we also present new data on the effects of the SLC38A9 mutants (**new Supplementary Figure S9** and detailed in point #4 below). SLC38A9 mutants, including SLC38A9 (1-119aa) activate FLCN by disrupting the lysosomal FLCN complex (LFC), similar to FLCN^{F118D}(PMID: 32868926). Although weaker than the effects of exogenous FLCN/FNIP2 transfection, SLC38A9 partially activates TFEB phosphorylation and decreases TFEB nuclear translocation (though not reaching statistical significance), presumably by activating the low levels of FLCN: FNIP2 present on the lysosome in *TSC2* KO cells. More detailed analyses will reveal if the FLCN: FNIP2 complex retains sensitivity to upstream inputs and/or the nature of post-translational modifications regulating its function in *TSC2* KO cells.

We have added this information to the Results (lines 402-459) and Discussion (515-543) of the manuscript.

2) In addition, what is the kinetics of the process? What is first in the tumorigenic

process upon TSC loss, mTORC1 hyperactivation, or TFEB nuclear translocation?

This is an excellent question but a difficult one to answer definitively *in vivo*. We examined expression of TSC2, TFEB and TFEB by immunohistochemistry in *PAX8-Cre; Tsc2 fl/wt* mice at 3 months. In **Response Figure #2** below, at 3 months of age (upper panel), only very focal areas of TSC2 protein loss (confirming focal bi-allelic *Tsc2* loss) are observed in morphologically unremarkable individual renal tubules (arrows) and small cysts (arrowhead) which are precursors to the later developing tumors. However, there is already increased nuclear TFEB and TFE3 apparent in these tubules (arrows) and cysts (arrowheads), suggesting nearly simultaneous

Response Figure 2: Immunostaining for TSC2, TFEB and TFE3 in *PAX8-Cre; Tsc2 fl/wt* mice at 3 months (upper panel) and 18 months (lower panel).

activation of TFEB and TFE3 with *Tsc2* loss. By 18 months of age (lower panel), full-blown cystadenomas and cystadenocarcinomas are evident, with *Tsc2* protein loss (confirming bi-allelic *Tsc2* inactivation) and nuclear TFEB/TFE3 accumulation. The presence of morphologically normal renal tubules with TSC2 loss and concurrent nuclear TFEB/TFE3 accumulation suggests that mTORC1 activation and TFEB nuclear translocation are very early events, preceding morphologic evidence of tumorigenesis. However, because these events occur so early, we cannot distinguish which comes first *in vivo*.

3) Also, these findings may represent an opportunity to identify novel therapeutics to treat tumors driven by the mTORC1/TFEB/TFE3 pathway. However, the authors stopped in the partially solved mechanism and did not test the effects of rapamycin and SLC38A9N/FLCNF118D mutants on TSC2 null-driven tumors. The manuscript is potentially interesting for Nature Communications. However, all these questions need to be addressed before considering the publication in this journal.

Thank you for this comment. We have now performed xenograft experiments with rapamycin and SLC38A9 (1-119aa)/ FLCN^{F118D} mutants as suggested. Please see detailed response to Point #2 and Point #3 below.

Some other comments:

1) Perhaps mTORC1 hyperactivation regulates the activity of SLC38A9, FLCN, or both by phosphorylation. Are these proteins phosphorylated by mTOR (since the mechanism is rapamycin-sensitive)? What residues are involved?

A limited number of previous studies have demonstrated that certain phosphorylation sites in FLCN (S62, S73, S302) are differentially regulated by the Tsc2-mTOR pathway (PMID:19695222, 21659605). However, the identity of the kinase as well functional significance in amino acid signaling to mTORC1 is not known. In the case of FNIP2, we did observe a mobility shift in Lamtor1/RagB immune-precipitates in starved *TSC2* KO cells, (**new Figure 7A,B, new Supplementary Figure S10D**), suggestive of a change in phosphorylation. An extensive phospho-proteomic characterization of the FLCN: FNIP2 complex, identification of the putative kinase, validation of phospho-sites (generation of phospho-specific antibodies and site mutants, *in vitro* kinase assays) and functional significance would be of significant interest and the subject of a future study.

We have added this to the Discussion section of the manuscript on lines 537-543.

2) The authors must test their hypothesis reverting tumor growth (xenograft) by expressing SLC38A9N and FLCNF118D mutants rescuing TFEB/TFE3 hypo phosphorylation and nuclear translocation.

In response to this question, we examined tumor xenograft growth in the following groups of mutants:

- 1) a) *TSC2* KO cells expressing empty vector
- b) SLC38A9 (1-119aa) (N-terminal domain)
- c) SLC38A9 Δ 110 lacking the N-terminal domain

We did not detect significant alterations in tumor growth between these groups (**new Supplementary Figure S9D**).

- 2) a) *TSC2* KO FLCN shRNA control cells
- b) *TSC2* KO FLCN shRNA cells stably expressing FLCN^{R164A}
- c) *TSC2* KO FLCN shRNA cells stably expressing FLCN^{F118D}
- d) *TSC2* KO FLCN shRNA cells stably expressing FLCN^{F118D}+ WT FNIP2

Tumor growth was partially suppressed in cells stably expressing FLCN^{F118D} and FLCN^{F118D}/FNIP2, however, these results were not significant (**new Supplementary Figure S10E, F**). While the reasons for insignificant tumor xenograft growth suppression by the FLCN^{F118D} and SLC38A9 (1-119aa) mutants remain unclear, it is notable that both activate FLCN by destabilizing the lysosomal FLCN complex, however neither acts to increase lysosomal localization of FLCN: FNIP2 which appears to be defective in *TSC2* KO cells.

We have added this to the Results section of the manuscript on lines 396-398 and 451-455.

3) Similar experiments of point 2 must be done using rapamycin. A comparison with torin1 must be done in parallel. It could be a great proof of principle for future therapy.

As requested, we have now performed a more extensive characterization of rapamycin and torin-treated cells and xenografts. We first examined the subcellular localization of TFEB and TFE3 in HEK293T WT and *TSC2* KO cells in response to treatment with the allosteric mTORC1 inhibitor rapamycin, as well as the ATP-competitive dual mTORC1/2 kinase inhibitor torin, by immunoblotting of nuclear/cytosolic fractions. Rapamycin significantly suppressed the nuclear localization of TFEB/TFE3 in *TSC2* KO cells, while torin had a more modest effect (**new Figure 2 E, F, G**). These findings are consistent with the fact that torin is a more complete mTORC1 inhibitor, which directly inhibits mTORC1-mediated phosphorylation of TFEB, while rapamycin is an incomplete mTORC1 inhibitor, with activity confined to a subset of mTORC1-substrates. Accordingly, previous work in *TSC2*-intact cells has shown that rapamycin does not directly inhibit mTORC1-mediated TFEB phosphorylation (PMID: 22343943, Figure 3A) due to its incomplete suppression of mTORC1 signaling.

In addition to these *in vitro* experiments, we also examined tumor xenograft growth in WT and *TSC2* KO cells in response to treatment with rapamycin and torin. Rapamycin significantly decreased tumor growth in both WT and *TSC2* KO xenografts (new **Figure 2K**), consistent with decreased activity of TFEB due to increased phosphorylation (new **Supplementary Figure S3B**). In contrast, torin did not significantly affect tumor growth (new **Figure 2K**), despite a near-similar reduction in phosphorylation of classic mTORC1 substrates (p-4E-BP1 and p-p70S6K) in *TSC2* KO tumor lysates (new **Figure 2L** and new **Supplementary Figure S3A**). In contrast to short-term *in vitro* torin treatment (**Figure 5A-C**), long-term torin treatment of xenografts did demonstrate a weak and partial rescue of TFEB phosphorylation in *TSC2* KO cells, as well as a parallel increase in p-AKT (as has previously been documented; PMID: 28757207). Notably, our previous studies in *Tsc1* KO primary keratinocytes (PMID: 31527310) and *TSC2*-null TRI-102 cells (PMID: 35072947), have shown that long term torin (>24 hrs) results in decreased TFE3 nuclear localization and downstream target expression (eg, GPNMB protein expression), respectively.

Taken together, these new experiments further support **our current model**. In the context of *TSC2* loss, rapamycin is sufficient to alleviate the mTORC1-mediated negative feedback loop inhibiting TFEB recruitment to the lysosome (via FLCN: FNIP2 relocalization, see point #7 below), without inhibiting mTORC1 activity towards TFEB, thus leading to paradoxically increased TFEB phosphorylation. In xenografts, this inactivation of TFEB with rapamycin leads to decreased xenograft growth. Short-term torin has similar effects as rapamycin on FLCN: FNIP2 localization but also potently inhibits mTORC1 activity towards TFEB, thus failing to rescue TFEB phosphorylation in the context of *TSC2* KO. Long-term torin (in xenografts) appears to behave somewhat between short term torin and rapamycin vis a vis TFEB, perhaps due to signaling feedback or other mechanisms, with partial rescue of p-TFEB levels in the context of *TSC2* loss, but this is insufficient to impact tumor growth.

We have added this to the Results section of the manuscript on lines 190-193, 199-206, and the Discussion on lines 496-498 and 528-553.

4) IFs of TFEB/TFE3 subcellular localizations are missing in SLC38A and FLCN mutant overexpressing cells.

To allow for more robust quantification, we examined the subcellular localization of TFEB and TFE3 in the following mutants by immunoblotting of nuclear-cytosolic fractions:

1) a) WT and b) *TSC2* KO cells expressing empty vector c) SLC38A9 (1-119aa) with stable expression of the N-terminal domain, d) SLC38A9 Δ 110 expressing cells lacking the N-terminal domain. Expression of SLC38A9 (1-119aa) in *TSC2* KO cells decreased the nuclear localization of TFEB and TFE3; however, these results were not statistically significant. SLC38A9 Δ 110 had no effect of TFEB/TFE3 subcellular localization as expected (**new Supplementary Figure S9C**).

2) WT and *TSC2* KO cells depleted for FLCN with shRNA, and stably expressing the following: a) WT FLCN, or FLCN mutants with b) abolished (FLCN^{R164A}), or c) uninhibited (FLCN^{F118D}) RagC GAP activity, in the absence/presence of transient FNIP2 expression. While expression of FLCN^{R164A} in *TSC2* KO cells did not further enhance TFEB/TFE3 nuclear localization, expression of FLCN^{F118D} partially suppressed nuclear localization of TFEB/TFE3, and this effect was further enhanced by transient co-expression of FNIP2 in these cells (**new Figure 7E, F**).

We have added this to the Results section of the manuscript on lines 393-396 and 442-447.

5) FLCN KO cells and tissues present an aberrant nuclear localization of TFEB. Also, the authors claim that “rapamycin-induced phosphorylation of TFEB is dependent on FLCN, similar to phosphorylation of TFEB in WT cells”. This claim is based on figure 7A, where the authors said that stable depletion of FLCN partially inhibited rapamycin-induced TFEB phosphorylation in TSC2 KO cells. However, the total levels of TFEB are highly reduced on this blot, making the ratio of P-TFEB/Total TFEB very similar to the effects of rapamycin alone in TSC ko cells. It may suggest that rapamycin and FLCN are not in the same pathway. How do the authors explain this? Perhaps, the same experiment but looking at the subcellular localization of TFEB/TFE3 may help to clarify this question.

We agree that the apparent changes in total TFEB levels with hypo-phosphorylation are disconcerting and we have spent a great deal of effort in trying to better understand the contributors. In characterizing the phosphorylation of TFEB in Figure 7A (which has now been replaced by **new Figure 7D, new Supplementary Figure S10A**), and throughout the manuscript, we have used specific, validated antibodies from Cell Signaling Technology (Phospho-TFEB (Ser211), #37681 and Phospho-TFEB (Ser122), #87932). In optimizing western blotting conditions for these antibodies, we were able to optimally detect p-TFEB using a mild lysis buffer from Cell Signaling Technology (Cell Lysis Buffer (10X) #9803), while p-TFEB was not or very faintly detected during lysis under more stringent lysis conditions (e.g. RIPA or Urea). Unfortunately, these mild lysis conditions likely favor cytosolic lysis with incomplete nuclear lysis, thus proteins with a predominant nuclear localization are artifactually under-represented in immunoblots. This is evidenced in Response Figure #1, above, which shows markedly reduced pull-down of TFEB in lysates of *TSC2* KO cells when using a mild buffer (NDLB), in contrast this is much less apparent for RIPA. In this way, we believe conditions favoring TFEB hypo-phosphorylation and nuclear localization appear to decrease total TFEB levels as well because it is sequestered in poorly lysed nuclei. Another contributing factor may be changes in electrophoretic mobility of TFEB due to phosphorylation changes which tend to resolve bands that were previously clustered together and may also give the appearance of decreased total protein levels. Similar results have been seen in a number of previously published studies:

- 1) PMID: 28055300 Supplemental Figure 1A
- 2) PMID:34597140, Figure 6D, lane 1-2
- 3) PMID: 22692423 Figure 3D (lysate) lane 1 vs lane 4
- 4) PMID: 33163944 Figure 3A, lane 1-2

Nevertheless, to ensure rigor, we have used multiple orthogonal techniques throughout the manuscript to examine TFEB/TFE3 subcellular localization and transcriptional activity in order to validate or confirm the p-TFEB immunoblots. In *TSC2* KO cells and xenografts, increased TFEB/TFE3 nuclear localization was confirmed by immunofluorescence (**new Figure 2A-D, 2I-J, new Supplementary Figure S2A-D**) and immunoblotting of nuclear-cytoplasmic fractions (**new Figure 2E-G, new Figure 7E-F, new Supplementary Figure S9C**- a total of nine new replicates) while increased TFEB/TFE3 transcriptional activity was confirmed by qRT-PCR/immunoblotting of lysosomal genes/proteins (**new Figure 1C-E, 5D**), RNASeq (**new Figure 1B, new Figure 4D**) and 4XCLEAR promoter activity(**new Figure 2H**). Furthermore, increased TFEB/TFE3 nuclear localization was also confirmed in 3 different transgenic mouse models (**new Figure 3**). Similarly, our new mechanistic data on the role played by FLCN: FNIP2 in regulating TFEB phosphorylation in *TSC2* KO cells (**new Figure 7D, new Supplementary Figure S10A**), is also supplemented by immunoblotting for changes in TFEB/TFE3 subcellular localization (**new Figure 7E-F**).

Moreover, and as detailed above, we now show that the lysosomal recruitment of both FLCN and FNIP2 upon amino acid starvation is significantly compromised in HEK293T *TSC2* KO cells (**new Figure 7A**), and is completely reversible upon mTORC1 inhibition with rapamycin or torin (**new Figure 7B-C, new Supplementary Figure S10D**), thus confirming that rapamycin and FLCN/FNIP2 do indeed act on the same pathway to regulate TFEB.

Also, using the same experimental strategy, what are the effects of TSC depletion in FLCN KO cells?

The question regarding the effects of *TSC2* depletion in the setting of *FLCN* loss is an interesting one. Stable deletion of *FLCN* in HEK293T *TSC2* KO cells did not further de-phosphorylate (already hypophosphorylated) TFEB (**new Supplementary Figure S10A**). However, while we were writing this manuscript, a 2021 study (PMID: 34253722) showed that transient, simultaneous deletion of *TSC2* and *FLCN* in Hela cells synergistically increased TFEB nuclear localization and activity. This indicates that the effects of combined *TSC2/FLCN* loss are likely contextual and cell-type specific.

We have added this to the Results section of the manuscript on lines 404-405.

6) The message of the paper may improve a lot with a scheme of the pathway identified

Thank you for this suggestion. We have now added a description of our current working model to the Discussion on lines 528-537.

7) There are very recent high-impact articles, even in the big brother Nature, around the pathway studied here. It would be great for general readers to compare your most significant findings and claims with the observations of these articles. Some examples: a) There are contradictory observations regarding the effects of FLCN loss and TSC loss on TFEB nuclear localization. b) Depletion of TFEB in a kidney-specific mouse model of BHD syndrome fully rescued the disease phenotype and associated lethality and normalized mTORC1 activity, whereas in the cancer models studied in this manuscript the double depletion of TFEB and TFE3 is required to reduce growth.

These are important points that we have now addressed more clearly in the Discussion and below.

- a) In Ballabio's critical 2020 study (PMID:32612235), siRNA-mediated depletion of TSC2 in HeLa cells did not affect phosphorylation or subcellular localization of TFEB. In stark contrast, we report that TSC2 loss in HEK293T cells and xenografts, as well as murine models of TSC2 loss-mediated renal tumorigenesis paradoxically leads to hypo-phosphorylation and nuclear translocation of TFEB. The reasons for these discordant findings are likely due to transient siRNA-mediated (Ballabio) vs permanent CRISPR-mediated (our study) TSC2 loss.
- b) There are likely context-specific factors that affect functional redundancy between MiT/TFE family members. Previous studies in other contexts have suggested that functional redundancy between TFEB and TFE3 is significant (PMID: 27298091, 27171064), though this was notably not observed in Ballabio's recent study (PMID:32612235) where *Tfeb* KO alone was sufficient to attenuate renal tumorigenesis. However, in the context of TSC2 specifically, we have performed growth assays and xenograft experiments (**new Figure 4B, C, Supplementary Figure S4**), and found that single gene inactivation of *TFEB* or *TFE3* had no effect on growth of *TSC2* KO xenografts, while KO of TFEB and TFE3 had a significant effect. In order to further characterize potential functional redundancy of TFEB and TFE3 in the *TSC2* KO background, we have now performed RNAseq on tumor xenograft specimens isolated from WT, *TSC2* KO, *TFEB* KO, *TFE3* KO and double *TFEB/TFE3* KO mice (**new Figure 4D**). Multiple, *TFEB*- and/or *TFE3*-regulated lysosomal gene sets were positively enriched by GSEA in *TSC2* KO compared to WT xenografts, validating our initial *in vitro* findings. While *TFEB/TFE3* DKO xenografts on the *TSC2* KO background showed significant, negative enrichment of all lysosomal gene sets compared to *TSC2* KO xenografts, findings with individual knockout of TFEB or TFE3 were not as strong or significant. These results are consistent with our observed effects on tumor growth and confirm redundancy among MiT/TFE family members in specific context of TSC2 loss.

We have added this information to the Results (lines 240-251) and Discussion (553-569) section of the manuscript.

Minor:

1) Some English editing is needed. The abstract and introduction are difficult to understand for general readers. There are many difficult to read sentences

The language in the abstract and introduction has now been simplified to make it easier to understand for the general reader.

2) What was the rationale to use rapamycin? did you perform a survey of several drugs, or its selection was by chance? An explanation would be useful in the manuscript.

Rapamycin is the best-characterized mTORC1-specific inhibitor available and the only one with family members currently approved for clinical use (eg, sirolimus). While a number of ATP-competitive mTOR kinase inhibitors exist (eg, torin), these are all dual mTORC1- and mTORC2-inhibitors, which complicates interpretation of the results. Given that rapamycin cleanly reversed TFEB hypo-phosphorylation in *TSC2* KO cells, this result was consistent with an mTORC1-

specific mechanism and became the focus of our study. However, we have done additional comparisons to torin in the revised study, including **new Figures 2E-G, 2K-L, 5C, 5E, 6A, 7B-C**, among others.

We have added this information to the Results (lines 188-193, 199-206, 297-299, 307-309, 316-318, 352-355, 425-427) and Discussion (496-498, 548-553) section of the manuscript.

Reviewer: 3 (Remarks to the Author):

In their manuscript titled “An mTORC1-mediated negative feedback loop constrains amino acid induced SLC38A9 activity in tuberous sclerosis complex”, Asrani and colleagues aim to provide a deeper understanding of how transcription factor EB (TFEB) driven lysosomal biogenesis can be induced as a compensatory process upon constitutive activation of mTORC1. The core idea of the manuscript is interesting, and the authors show that due to deficient recruitment of TFEB and TFE3 to the lysosomes, they escape their inhibiting phosphorylation, which allow them to translocate to the nucleus and activate their target genes. Those data, although not without merit, do not generate a convincingly novel concept as neither, the precise mechanism behind the defective recruitment, nor the link to amino acid metabolism or tuberous sclerosis has sufficiently been addressed. To improve the quality and increase the novelty of their finding, the authors should perform more experiments to better understand how TFEB can escape the inhibitory phosphorylation that is catalyzed upon short term activation of mTOR.

Major comments

1) The title is misleading, as the main focus of the paper is not to link the TFEB-driven negative feedback of constitutive mTORC1 activation to tuberous sclerosis. The authors might consider to either tone down the title, or to perform further experiments with a clearer focus on disease progression and/or amelioration.

We have modified the title to redirect the focus away from tuberous sclerosis complex as requested. The new title is: “An mTORC1-mediated negative feedback loop constrains amino acid-induced FLCN-Rag activation downstream of TSC2 loss”

2) The authors show that stable loss of TSC2 promotes TFEB mediated transcription of lysosomal genes. However, although they have obtained TSC1 KO, as well as TSC1/2 double KO cells, they do not confirm their gene expression and protein abundance findings (Fig1) in those cell lines. It would be an important control to provide those panels, as the authors have also done for the nuclear localization IF and WB.

Thank you for the comment. In new experiments, we have now examined the expression of multiple lysosomal CLEAR gene transcripts (**new Figure 1C, and new Supplementary Figure S1A**) and lysosomal proteins by immunoblotting (**new Figure 1D**) in HEK293T WT, *TSC1* KO, *TSC2* KO and dual *TSC1/2* KO cells. The majority of lysosomal genes and proteins were significantly enriched in *TSC2* and *TSC1/2* KO cells, compared to WT cells. Interestingly, while the data for *TSC1* KO cells trends in the same direction as *TSC2* and *TSC1/2* KO cells in most experiments, differences between *TSC1* KO and WT cells are not statistically significant. The reasons for the weaker phenotype in *TSC1* KO HEK293T cells are unclear since nuclear TFEB and TFE3 accumulate in renal cystadenomas from a murine model of *Tsc1* KO (**new Figure 3D**), similar to levels seen in similar models of *Tsc2* loss (**Figure 3A-C**). In addition, the clear rescue

of all phenotypes observed in *TSC2* KO HEK293T cells with rapamycin provides strong evidence that our findings are mediated by mTORC1. To provide additional rigor, we have now also performed RNAseq on tumor xenograft specimens isolated from WT and *TSC2* KO mice (**new Figure 4D**). Multiple, *TFEB*- and/or *TFE3*-regulated lysosomal gene sets were positively enriched by GSEA in *TSC2* KO compared to WT xenografts, validating our initial *in vitro* findings in HEK293T cells.

We have added this information to the Results section of the manuscript (lines 149-154 and 245-247).

3) Why is the TFEB-driven transcriptional increase not observed in normal kidney cells although other non-cancerous cells (e.g., primary keratinocytes) manifest that too? Is it restricted to rapidly proliferating cells?

The increase in TFEB activity is restricted to the cells that have bi-allelic *Tsc2* loss. Importantly, the normal kidney surrounding cystadenomas in the A/J and Pax8-cre models (**Figure 3A and 3C**) does not have *Tsc2* protein loss since in both models there is only one-copy (mono-allelic) loss of *Tsc2* at baseline, with a second, spontaneous “hit” restricted to the cystadenoma lesions or the tubules immediately surrounding these lesions. This is best visualized with *Tsc2* immunostaining which is now performed for the Pax8-cre model in **Figure 3C** and **Response Figure 2 above**) and for the A/J model in a recent publication from our group (PMID: 35072947). Thus, the increase in TFEB/TFE3 nuclear accumulation (and presumably activity) is restricted to the cells with *Tsc2* bi-allelic loss and consequent mTORC1 constitutive activation.

We have added this information to the Results section of the manuscript on lines 163-164 and 219-221.

4) Because of the difference in inhibitory potential between rapamycin and torin1 (especially with respect to the kinase function), it would be interesting, to address the localization of TFEB, as well as the transcriptional activation of its targets following torin1 treatment. Those results might also help to better characterize the precise mechanism of how constitutive mTORC1 inhibition leads to an increase in TFEB phosphorylation. Have the authors tried similar experiments?

As requested, we have now performed a more extensive characterization of rapamycin and torin-treated cells and xenografts. We first examined the subcellular localization of TFEB and TFE3 in HEK293T WT and *TSC2* KO cells in response to treatment with the allosteric mTORC1 inhibitor rapamycin, as well as the ATP-competitive dual mTORC1/2 kinase inhibitor torin, by immunoblotting of nuclear/cytosolic fractions. Rapamycin significantly suppressed the nuclear localization of TFEB/TFE3 in *TSC2* KO cells, while torin had a more modest effect (**new Figure 2 E, F, G**). These findings are consistent with the fact that torin is a more complete mTORC1 inhibitor, which directly inhibits mTORC1-mediated phosphorylation of TFEB, while rapamycin is an incomplete mTORC1 inhibitor, with activity confined to a subset of mTORC1-substrates. Accordingly, previous work in *TSC2*-intact cells has shown that rapamycin does not directly inhibit mTORC1-mediated TFEB phosphorylation (22343943, Figure 3A) due to its incomplete suppression of mTORC1 signaling.

In addition to these *in vitro* experiments, we also examined tumor xenograft growth in WT and *TSC2* KO cells in response to treatment with rapamycin and torin. Rapamycin significantly

decreased tumor growth in both WT and *TSC2* KO xenografts (new **Figure 2K**), consistent with decreased activity of TFEB/TFE3 due to increased phosphorylation (new **Supplementary Figure S3B**). In contrast, torin did not significantly affect tumor growth (new **Figure 2K**), despite a near-similar reduction in phosphorylation of classic mTORC1 substrates (p-4E-BP1 and p-p70S6K) in *TSC2* KO tumor lysates (new **Figure 2L** and new **Supplementary Figure S3A**). In contrast to short-term *in vitro* torin treatment (**Figure 5A-C**), long-term torin treatment of xenografts did demonstrate a weak and partial rescue of TFEB phosphorylation in *TSC2* KO cells, as well as a parallel increase in p-AKT (as has previously been documented; PMID: 28757207). Notably, our previous studies in *Tsc1* KO primary keratinocytes (PMID: 31527310) and *TSC2*-null TRI-102 cells (PMID: 35072947), have shown that long term torin (>24 hrs) results in decreased TFE3 nuclear localization and downstream target expression (eg, GPNMB protein expression), respectively.

Taken together, these new experiments further support **our current model**. In the context of *TSC2* loss, rapamycin is sufficient to alleviate the mTORC1-mediated negative feedback loop inhibiting TFEB recruitment to the lysosome (via FLCN: FNIP2 relocalization, see point #7 below), without inhibiting mTORC1 activity towards TFEB, thus leading to paradoxically increased TFEB phosphorylation. In xenografts, this inactivation of TFEB with rapamycin leads to decreased xenograft growth. Short-term torin has similar effects as rapamycin on FLCN: FNIP2 localization but also potently inhibits mTORC1 activity towards TFEB, thus failing to rescue TFEB phosphorylation in the context of *TSC2* KO. Long-term torin (in xenografts) appears to behave somewhat between short term torin and rapamycin vis a vis TFEB, perhaps due to signaling feedback or other mechanisms, with partial rescue of p-TFEB levels in the context of *TSC2* loss, but this is insufficient to impact tumor growth.

We have added this information to the Results section of the manuscript on lines 190-193, 199-206, and the Discussion on lines 496-498 and 528-553.

5) From the WB in Suppl. Fig2D and E it seems more as if total TFEB decreases in TCS KO cells. In absolute nuclear amounts, there seems to be no difference between WT and KO cells. Would not similar levels of nuclear TFEB result in similar degrees of gene expression activation independently of the amount of cytoplasmic TFEB?

We agree that in old Supplementary Figure S2D (now new **Supplementary Figure S2F**), absolute nuclear TFEB levels do not appear to be increased in *TSC2/TSC1,2* KO cells. This may be partially an artifact of nuclear-cytoplasmic fractionation experiments, thus the nuclear to cytoplasmic ratio (rather than the absolute nuclear level) is typically the accepted readout for these assays. In support of the functional significance of increased nuclear to cytoplasmic ratio of TFEB, TFEB-GFP was clearly hypo-phosphorylated (new **Supplementary Figure S7E**), and nuclear TFEB levels were elevated by immunofluorescence (new **Figure 2C**) in *TSC2* KO cells in these experiments. In the case of *Tsc2*-null, TTJ cells in old Supplementary Figure S2E (now new **Supplementary Figure S2G**), despite no differences in absolute nuclear TFEB levels *in vitro*, we were able to observe an elevated and quantifiable increase in TFEB nuclear localization (new **Figure 3A, B**) and transcriptional increase in CLEAR genes (new **Figure 1F, G**) and increased lysosomal proteins (new **Figure 1H**) in the parental renal tumors in *Tsc2* +/- A/J mice from which these cells were derived. In addition, *Tsc2*-null, TTJ cells also demonstrated elevated lysosomal protein expression (new **Figure 1I, J**).

To ensure rigor, we have used multiple orthogonal techniques to quantify TFEB/TFE3 nuclear localization and transcriptional activity in addition to examining p-TFEB levels. In *TSC2* KO cells and xenografts, increased TFEB/TFE3 nuclear localization was confirmed by

immunofluorescence (new Figure 2A-D, 2I-J, new Supplementary Figure S2A-D) and immunoblotting of nuclear-cytoplasmic fractions (new Figure 2E-G, new Figure 7E-F, new Supplementary Figure S9C- a total of nine new replicates) while increased TFEB/TFE3 transcriptional activity was confirmed by qRT-PCR/immunoblotting of lysosomal genes/proteins (new Figure 1C-E, 5E), RNASeq (new Figure 1B, new Figure 4D) and 4XCLEAR promoter activity (new Figure 2H). Furthermore, increased TFEB/TFE3 nuclear localization was also confirmed in 3 different transgenic mouse models (new Figure 3).

6) The KO experiments of TFEB and TFE3 are intriguing as it partially seems as if they (over-)compensate for one another (e.g., in the tumor growth experiment) while in other readouts (e.g., the WB in Fig4A) they seem to not always be able to compensate. A better characterization of the single vs the double knockouts would help to clarify those partially conflicting results.

The *TFEB/TFE3* KO growth assays and xenograft experiments (new Figure 4B, C, Supplementary Figure S4), are indeed intriguing and suggest an element of functional redundancy, as has been previously described for TFEB and TFE3 (PMID: 27298091, 27171064), though this was notably not observed in the recent *FLCN* study where *Tfeb* KO alone was sufficient to attenuate renal tumorigenesis (PMID: 32612235). In order to further characterize potential functional redundancy of TFEB and TFE3 in the *TSC2* KO background, we have now performed RNAseq on tumor xenograft specimens isolated from WT, *TSC2* KO, *TFEB* KO, *TFE3* KO and double *TFEB/TFE3* KO mice (new Figure 4D). Multiple, *TFEB*- and/or *TFE3*-regulated lysosomal gene sets were positively enriched by GSEA in *TSC2* KO compared to WT xenografts, validating our initial *in vitro* findings. While *TFEB/TFE3* DKO xenografts on the *TSC2* KO background showed significant, negative enrichment of all lysosomal gene sets compared to *TSC2* KO xenografts, findings with individual knockout of TFEB or TFE3 were not as strong or significant. These results are consistent with our observed effects on tumor growth and confirm redundancy among MiT/TFE family members in this context.

The *in vitro* results (new Figure 4A), also suggest that single *TFEB* or *TFE3* KO is sufficient to dampen the phosphorylation of direct mTORC1 substrates (p-4E-BP1 and p-P70S6K), and we further characterized these cells in search of a potential mechanism. Notably, a prior study (PMID: 28619945) has demonstrated that MiT/TFE activation and CLEAR-mediated gene transcription may lead to increased RagD expression (since Rag D contains a CLEAR element in its promoter) and thus contribute to increased mTORC1 signaling. Though this had not been previously shown for *TSC2* KO cells, we hypothesized that CLEAR activation downstream of *TSC2* loss may lead to increased RagD transcription and contribute to increased mTORC1 signaling in these cells. If this is the case, then TFE3/TFEB depletion in the setting of *TSC2* loss might in turn decrease RagD transcription and dampen mTORC1 signaling.

To begin to address this hypothesis, we first examined *TSC2* KO cells and xenografts and observed that, consistent with our finding of CLEAR activation in these cells, *RRAGD* mRNA and RagD protein levels were increased in *TSC2* KO and *TSC1, 2* KO cells and xenografts, compared to their WT counterparts (new Figure 1 C, D and Supplementary Figure S6A, B). In *TSC2* KO cells, we show that *RAGD* shRNA partially suppresses mTORC1 substrate phosphorylation by immunoblotting (new Supplementary Figure S6C), and overexpression of active RagD further boosted 4EBP1 phosphorylation in *TSC2* KO cells (new Figure 6B). These data suggest that there is at least some residual activity of RagD in *TSC2* KO cells *vis a vis* non-MiT/TFE mTORC1 substrates such as 4EBP1 and S6K.

Significantly, we find that *RAGD* transcription can be suppressed in *TSC2* KO cells with single *TFE3* or *TFEB* KO or *TFEB/TFE3* KO (**new Supplementary Figure S6D**), paralleling the diminished mTORC1 signaling we observed in single *TFE3* or *TFEB* KO or *TFEB/TFE3* KO cells *in vitro* (**new Figure 4A**). These results indicate that similar to what has been previously reported in WT cells, increased RagD transcription also potentially contributes to mTORC1 signaling in *TSC2* KO cells, and the resulting downregulation in *RAGD* transcription in the single KO clones is sufficient to dampen mTORC1 signaling. Further studies will be required to assess the contribution of *RRAGD* in driving tumor cell proliferation in the context of *TFEB/TFE3* deletion in cells with *TSC2* loss.

We have added this information to the Results (lines 240-251, lines 252-271; lines 330-332) and Discussion (553-565) section of the manuscript.

7) It would be of great value for the manuscript if the authors would try to address how loss of TCS could influence the interaction between TFEB and the Rag GTPases and how rapamycin would contribute to the reversal of the phenotype.

We have now provided additional mechanistic details underlying the regulation of TFEB hypo-phosphorylation in *TSC2* KO cells, and its reversal by mTORC1 inhibition. As previously described (PMID: 24081491, 24095279, 29848618, 31672913), multiple lines of evidence support a role for FLCN and its binding partners FNIP (the FLCN: FNIP2 complex) in coordinating cellular responses to amino acid availability via Rag heterodimer activation, and corresponding mTORC1 activation at the lysosome. In response to amino acid starvation, the FLCN: FNIP2 complex is first recruited to the lysosome, where it interacts with the Rag-Ragulator complex, and this lysosomal re-localization is an essential prerequisite for its role as a RagC/D GAP upon nutrient re-stimulation leading to subsequent mTORC1 activation.

We now demonstrate that the lysosomal recruitment of both FLCN and FNIP2 upon amino acid starvation is significantly compromised in HEK293T *TSC2* KO cells (**new Figure 7A**), and this is completely reversed upon mTORC1 inhibition with rapamycin or torin (**new Figure 7B, C, new Supplementary Figure S10 C, D**). Consistent with these *in vitro* findings, expression of FNIP2 was decreased *in vivo* in renal tumors in *Tsc2 +/-* mice by immunohistochemistry (**new Figure 7G**). Importantly, FLCN depletion (first four lanes of **new Supplementary Figure S10A**) or FNIP2 depletion (**new Supplementary Figure S10B**) by shRNA is sufficient to suppress TFEB phosphorylation in WT cells, indicating that both components of the dimer are required for TFEB inactivation. Taken together, these data suggested the hypothesis that failure of FLCN: FNIP2 to localize to the lysosome and activate the Rag heterodimer in *TSC2* KO cells could underlie TFEB hypo-phosphorylation in this context.

To test this, we examined whether lysosomal re-localization or activation of FLCN: FNIP2 could rescue TFEB phosphorylation in *TSC2* KO cells. It has been noted in previous studies that concurrent over-expression of FLCN and FNIP2 leads to constitutive lysosomal localization of FLCN, independent of nutrient conditions (PMID: 24081491, 18663353, 24095279, 27113757, 28039480). Corresponding to the observed defect in FLCN: FNIP2 recruitment to the lysosome in *TSC2* KO cells, co-expression of WT FLCN and FNIP2 fully rescued TFEB phosphorylation in *TSC2* KO cells (lanes 4 and 5 of **new Figure 7D**). To test the effects of activating or inactivating FLCN mutants, we did similar experiments in *TSC2* KO cells depleted for endogenous FLCN with shRNA. Co-transfection of FNIP2 with WT FLCN or activated FLCN^{F118D} (lanes 8-9 or 12-13 of **new Figure 7D**, lanes 9-13 of **Supplementary Figure S10A**) led to a rescue of p-TFEB, while a similar rescue was not seen with inactive FLCN^{R164A} as a negative control. To confirm the

functional significance of these results, we examined the subcellular localization of TFEB/TFE3 in cells expressing FLCN mutants. *TSC2* KO cells depleted for FLCN with shRNA, and stably expressing FLCN^{F118D} partially suppressed nuclear localization of TFEB/TFE3, and this effect was further enhanced by transient co-expression of FNIP2 in these cells (**new Figure 7E, F**).

Taken together, these data suggest that the FLCN:FNIP2 complex fails to be recruited to the lysosomal membrane during starvation in *TSC2* KO cells, and this is mTORC1-dependent and reversible by concurrent overexpression of FLCN and FNIP2 which restores these proteins to the lysosomal membrane. Consistent with this model, we also present new data on the effects of the SLC38A9 mutants (**new Supplementary Figure S9**). SLC38A9 mutants, including SLC38A9 (1-119aa) activate FLCN by disrupting the lysosomal FLCN complex (LFC), similar to FLCN^{F118D}. Although weaker than the effects of exogenous FLCN/FNIP2 transfection, SLC38A9 partially activates TFEB phosphorylation and decreases TFEB nuclear translocation (though not reaching statistical significance), presumably by activating the low levels of FLCN: FNIP2 present on the lysosome in *TSC2* KO cells. More detailed analyses will reveal if the FLCN: FNIP2 complex retains sensitivity to upstream inputs and/or the nature of post-translational modifications regulating its function in *TSC2* KO cells.

We have added this information to the Results (lines 402-459) and Discussion (515-543) of the manuscript.

8) In line with point 7, while the authors nicely show how forced or artificial tethering of TFEB to the lysosomes results in its phosphorylation, whereas inhibition of TFEB going to the lysosomes results in its hypo-phosphorylation, it remains unclear, how the TFEB is kept away from the lysosomes upon lack of TFEB. It might be insightful to try to pull down TFEB in cells proficient or deficient for TSC and potentially identify different interaction partners that could help to elucidate the molecular mechanism.

Thank you for the comment. In response to amino acid starvation, the FLCN: FNIP2 complex is first recruited to the lysosome, where it interacts with the Rag-Ragulator complex, and this lysosomal re-localization is an essential prerequisite for its role as a RagC/D GAP. TFEB is consequently recruited to the lysosome for phosphorylation by mTORC1 specifically in response to amino acid stimulation, via activated RagC/D downstream of folliculin (FLCN) activation, as was demonstrated in Ballabio's critical 2020 study (PMID:32612235). We previously showed that forced tethering of TFEB rescues its phosphorylation in *TSC2* KO cells. We now show (as described in detail in point#7, above) that in *TSC2* KO cells, the FLCN:FNIP2 complex fails to be recruited to the lysosomal membrane during starvation, and this is mTORC1-dependent and reversible by concurrent overexpression of FLCN and FNIP2 (but not either one alone), which restores both these proteins to the lysosomal membrane. As a result, TFEB lysosomal recruitment and phosphorylation and reversal of nuclear localization were fully rescued by FLCN: FNIP2 combined overexpression in *TSC2* KO cells.

We have added this information to the Results (lines 402-459) and Discussion (515-543) of the manuscript.

We agree that a comprehensive characterization of the TFEB phosphor-proteome/ interactome in *TSC2* KO cells would also be of significant interest, to validate existing phospho-sites and/or identify new phospho-sites (for e.g. phosphorylation of potential residues in the N-terminal domain of TFEB that regulate binding to Rag GTPases) or novel/differentially regulated binding partners.

9) Similarly, the involvement of FLCN, LFC and SLC38A9 would require further

experiments. Also here, identification of the interactome of FLCN and SLC38A9 following knockout of TSC might help to better understand their involvement. Alternatively, the involvement of SLC38A9 could be demonstrated by using specific antagonists and subsequently addressing mTORC1 activity in general as well as with respect to TFEB.

A limited number of previous studies have demonstrated that certain phosphorylation sites in FLCN (S62, S73, S302) are differentially regulated by the Tsc2-mTOR pathway (PMID:19695222, 21659605). However, the identity of the kinase as well functional significance in amino acid signaling to mTORC1 is not known. In the case of FNIP2, we did observe a mobility shift in Lamtor1/RagB immune-precipitates in starved *TSC2* KO cells, (**new Figure 7A, new Supplementary Figure S10D**), suggestive of a change in phosphorylation. An extensive phospho-proteomic characterization of the FLCN:FNIP2 complex, identification of the putative kinase, validation of phospho-sites (generation of phospho-specific antibodies and site mutants, *in vitro* kinase assays) and functional significance would be of significant interest and the subject of a future study.

We have added this to the Discussion section of the manuscript on lines 537-543.

Minor comments

1) Revise sentence “We demonstrate the Rag-mediated TFEB phosphorylation is in response to amino acids is..”; page 5

This has now been corrected.

2) The DAPI staining in Fig. 2B seems overexpose

These images have now been replaced.

3) The parenthesis “(hereafter referred to asTCS2 KO and WT..)” should be moved to the first results paragraph, as the respective abbreviations have already been used there.

This statement has now been moved to the first paragraph.

REVIEWER COMMENTS

Reviewer #1 (Remarks to the Author):

While the authors acknowledge 3 key papers in their rebuttal, the misattribution of previous discoveries in their manuscript is somewhat unfortunate (constitutive TFEB nuclear localization, TFEB hypophosphorylation, paradoxical rapamycin-induced phosphorylation and cytoplasmic localization, and impaired regulation with N-terminal deletion), but this should be no reason to hold the manuscript back and the authors have otherwise done an outstanding job addressing my queries.

Reviewer #2 (Remarks to the Author):

The authors made a big effort to improve and answer some questions asked during the revision of this manuscript. However, the authors did not provide more mechanistic studies about the contribution of SLC38A9 in this pathway and how it signals to the FLCN:FNIP2 complex. SLC38A9 partially activates TFEB phosphorylation and decreases TFEB nuclear translocation but does not reach statistical significance. Most importantly, the expression of SLC38A9N does not affect xenograft growth. Moreover, whether and how SLC38A9 and FLCN are modulated by mTORC1 or other kinases has not been addressed. I think that the contribution of SLC38A9 is still not clear in this manuscript version. I suggest providing more mechanistic and direct experimental evidence of the link between SLC38A9 with the FLCN: FNIP2 pathway or excluding SLC38A9 from the study.

Reviewer #3 (Remarks to the Author):

Through extensive revision, K. Asrani and colleagues significantly increased the quality of their manuscript and added important and interesting new insights into how MiT/TFEB hyperactivation is mediated in TSC2 deficient cells.

In detail, the title has been adapted accordingly.

In addition, various experiments, such as the characterization of TSC1, TSC2 and TSC1/2 double KO cells, a more thorough comparison of the effects of Torin 1 and rapamycin on TFEB localization, extensive

prove of changed nuclear localization of TFEB with different methods have been added to the manuscript.

Importantly, the authors have put further effort into understanding the hypophosphorylation and subsequent nuclear localization of TFEB observed in TSC2 deficient cells. They could identify a decrease in FLCN:FNIP2 lysosomal localization to be causative for the reduction in phosphorylation.

Reviewer(s)' Comments to Author:

Reviewer: 1 (Remarks to the Author):

While the authors acknowledge 3 key papers in their rebuttal, the misattribution of previous discoveries in their manuscript is somewhat unfortunate (constitutive TFEB nuclear localization, TFEB hypophosphorylation, paradoxical rapamycin-induced phosphorylation and cytoplasmic localization, and impaired regulation with N-terminal deletion), but this should be no reason to hold the manuscript back and the authors have otherwise done an outstanding job addressing my queries.

We thank the reviewer for this thoughtful appraisal of our revisions. We apologize for the misattribution of previous findings by other groups, and have attempted to rectify the citations as appropriate, as indicated below, and highlighted in the manuscript:

- 1) Constitutive TFEB nuclear localization: lines 97-100
(Citations added: 34253722, 21804531)
- 2) TFEB hypo-phosphorylation:
-lines 86-89 (Citations added: 28055300)
-lines 276-278 (Citations added: 28055300, 22343943, 22576015)
- 3) Impaired regulation with N-terminal deletion: lines 346-352
(Citations added: 22692423, 32612235, 23401004, 29507340)

Reviewer: 2 (Remarks to the Author):

The authors made a big effort to improve and answer some questions asked during the revision of this manuscript. However, the authors did not provide more mechanistic studies about the contribution of SLC38A9 in this pathway and how it signals to the FLCN:FNIP2 complex. SLC38A9 partially activates TFEB phosphorylation and decreases TFEB nuclear translocation but does not reach statistical significance. Most importantly, the expression of SLC38A9N does not affect xenograft growth. Moreover, whether and how SLC38A9 and FLCN are modulated by mTORC1 or other kinases has not been addressed. I think that the contribution of SLC38A9 is still not clear in this manuscript version. I suggest providing more mechanistic and direct experimental evidence of the link between SLC38A9 with the FLCN: FNIP2 pathway or excluding SLC38A9 from the study.

Thank you for these comments. In our revision, we focused predominantly on elucidating additional mechanisms involving FLCN:FNIP2 complex activity and agree that we were unable to provide more mechanistic details regarding the contribution of SLC38A9. While we are unsure as to why expression of SLC38A9 mutants *in vitro* and in xenograft assays fell short of showing statistically significant differences, it is notable that the SLC38A9 mutants were expressed in parental *TSC2* KO cells, and it is possible that endogenous expression of SLC38A9 may have confounded these results. Similar to our experiments with the FLCN mutants in *FLCN*-depleted cells, we did attempt to generate these mutants in *SLC38A9*-depleted cells. However, the *TSC2* KO cells with *SLC38A9* depletion demonstrated poor viability, thus limiting our ability to study

these mutants. In light of these results, as suggested, we have now excluded SLC38A9 (Figure S9 in the previous version of the manuscript) from the entirety of our study.

Reviewer: 3 (Remarks to the Author):

Through extensive revision, K. Asrani and colleagues significantly increased the quality of their manuscript and added important and interesting new insights into how MiT/TFEB hyperactivation is mediated in TSC2 deficient cells.

In detail, the title has been adapted accordingly.

In addition, various experiments, such as the characterization of TSC1, TSC2 and TSC1/2 double KO cells, a more thorough comparison of the effects of Torin 1 and rapamycin on TFEB localization, extensive prove of changed nuclear localization of TFEB with different methods have been added to the manuscript.

Importantly, the authors have put further effort into understanding the hypophosphorylation and subsequent nuclear localization of TFEB observed in TSC2 deficient cells. They could identify a decrease in FLCN:FNIP2 lysosomal localization to be causative for the reduction in phosphorylation.

We thank the reviewer for this positive appraisal of our revision.